# *Clostridioides difficile* exploits toxin-mediated inflammation to alter the host nutritional landscape and exclude competitors from the gut microbiota

Joshua R. Fletcher[1], Colleen M. Pike [1], Ruth J. Parsons [1], Alissa J. Rivera[1], Matthew H. Foley[1],
Michael R. McLaren [1], Stephanie A. Montgomery [2] & Casey M. Theriot [1✉]

*Clostridioides difficile* is a bacterial pathogen that causes a range of clinical disease from mild to moderate diarrhea, pseudomembranous colitis, and toxic megacolon. Typically, *C. difficile* infections (CDIs) occur after antibiotic treatment, which alters the gut microbiota, decreasing colonization resistance against *C. difficile*. Disease is mediated by two large toxins and the expression of their genes is induced upon nutrient depletion via the alternative sigma factor TcdR. Here, we use tcdR mutants in two strains of *C. difficile* and omics to investigate how toxin-induced inflammation alters *C. difficile* metabolism, tissue gene expression and the gut microbiota, and to determine how inflammation by the host may be beneficial to *C. difficile*. We show that *C. difficile* metabolism is significantly different in the face of inflammation, with changes in many carbohydrate and amino acid uptake and utilization pathways. Host gene expression signatures suggest that degradation of collagen and other components of the extracellular matrix by matrix metalloproteinases is a major source of peptides and amino acids that supports *C. difficile* growth in vivo. Lastly, the inflammation induced by *C. difficile* toxin activity alters the gut microbiota, excluding members from the genus *Bacteroides* that are able to utilize the same essential nutrients released from collagen degradation.

[1] Department of Population Health and Pathobiology, College of Veterinary Medicine, North Carolina State University, 1060 William Moore Drive, Raleigh, NC 27607, USA. [2] Department of Pathology and Laboratory Medicine, Lineberger Comprehensive Cancer Center, University of North Carolina School of Medicine, Chapel Hill, NC, USA. ✉email: cmtherio@ncsu.edu

Clostridioides difficile (C. difficile) is a Gram-positive anaerobic gut pathogen that causes diarrhea, with severe cases resulting in significant morbidity and mortality[1]. C. difficile produces two large toxins, TcdA and TcdB, that glycosylate host Rho and Rac GTPases, leading to a disruption in the actin cytoskeleton and loss of epithelial barrier integrity; the subsequent apoptosis and tissue damage result in significant inflammation[2]. Recent work revealed inflammation can be beneficial for prominent enteric pathogens, such as Salmonella enterica and Vibrio cholerae, by providing a metabolic niche for them in the gut[3–5]. Whether inflammation can benefit C. difficile is not clear, yet patients with inflammatory bowel disease (IBD) are four times more likely to acquire C. difficile infection (CDI) compared to the general population, suggesting C. difficile may thrive in an inflamed environment[6–10]. CDI-mediated inflammation results in drastic shifts to the murine gut metabolome, with alterations in amino acid and peptide metabolite concentrations, indicating that toxin activity induces an altered gut metabolic profile[11]. Although a nutritional generalist, C. difficile is auxotrophic and requires multiple amino acids, including the branched-chain amino acids and proline that are used in Stickland metabolism for ATP production and regeneration of NAD+; thus, C. difficile must acquire these nutrients from its environment[12–15]. We therefore reasoned that C. difficile may gain access to these nutrients by exploiting the host inflammatory response. We hypothesized that toxin-mediated inflammation alters the host gut environment to benefit C. difficile growth and persistence, either through nutrient availability and/or the composition of the gut microbiota, potentially excluding competitors or selecting for allies[3,5].

We addressed this hypothesis by taking a holistic approach to define the response of the pathogen, the host, and the gut microbiota in the face of inflammation induced by C. difficile toxins. Here we show that the toxin-producing strain (wild type C. difficile) induces a unique C. difficile transcriptomic signature compared to the toxin null strain (isogenic tcdR mutant), indicating that inflammation shapes C. difficile metabolism in vivo. C. difficile transcripts for carbohydrate and branched-chain amino acid metabolism genes were differentially regulated in response to toxin-induced inflammation, which is a reflection of the nutrients available in the inflamed gut. Host tissue extracellular matrix (ECM)-degrading matrix metalloproteinase (MMP) transcripts, encoding enzymes responsible for breaking down amino acid rich collagen, were also increased in expression during peak inflammation. Additionally, we show that toxin activity leads to a reduction and reorganization in collagen around cells in vitro, which provided C. difficile a mechanism to acquire essential Stickland reaction substrates, supporting growth. Colonization with toxin-producing C. difficile also led to alterations in the gut microbial community structure, with inflammation suppressing the return of members from the Bacteroidaceae Family. Our results were conserved across different strains, as toxin activity of the epidemic C. difficile R20291 strain elicits similar responses in a mouse model, suggesting that these effects may be conserved across toxigenic C. difficile strains from phylogenetically distinct backgrounds.

## Results

### Wild type C. difficile induces significantly more inflammation and tissue damage than a tcdR mutant.
Antibiotic treated mice were challenged with $10^5$ spores of C. difficile 630Δerm (wild type, or wild type mice hereafter) or an isogenic C. difficile 630Δerm tcdR::ermB (tcdR, or tcdR mice) mutant on day 0 and clinical signs of disease were monitored for 4 days post-challenge (Fig. 1a). Another group of mice were given the antibiotic but

were not challenged with C. difficile (no C. diff, or uninfected mice). Mutation of tcdR has been reported to significantly reduce levels of both toxin gene expression and toxin protein production in vitro[34,35]. Nearly five-fold fewer tcdR vegetative cells were recovered in the feces relative to wild type at day 1 ($p = 0.0314$, Kruskal–Wallis with Dunn's correction for multiple comparisons); however, there was no difference by day 3, nor were significant differences detected in fecal spores between the two strains at either day (Fig. 1b, c). Cecal content from days 2 and 4 did not harbor significantly different vegetative cells or spores; however, nearly $10^5$-fold more toxin activity was detected in wild type mice compared to tcdR mice at both days, indicating that the tcdR mutant behaves similarly in vivo as it does in vitro with respect to toxin production (Supplementary Fig. 1a–c, Fig. 1d). Accordingly, histopathological analysis of cecal tissue found significantly increased inflammation in wild type mice when compared to uninfected controls (no C. diff) at day 2 ($p = 0.006$; Two-Way ANOVA with Tukey's multiple comparisons test), as well as significantly more epithelial damage when compared to both uninfected controls and tcdR mice at day 4 ($p = 0.024$ for both; Two-Way ANOVA with Tukey's multiple comparisons test) (Fig. 1e–f). While cecal inflammation, epithelial damage, and edema were lower in tcdR mice at day 2 relative to wild type mice, it was not statistically significant. Tissue damage in wild type mice was even more pronounced in colonic tissue (Supplementary Fig. 1d). Together, these data show that the tcdR mutant fails to produce much detectable toxin activity in vivo, and consequently does not elicit significant inflammatory damage to host gut tissue.

### Toxin-mediated inflammation significantly alters the C. difficile transcriptome in vivo.
As colonization with wild type C. difficile leads to significant increases in inflammation and damage to the cecal epithelium, we hypothesized that C. difficile would shift its transcriptome to reflect such dramatic differences in the inflammatory environment[36]. However, a tcdR mutant in the R20291 strain is pleiotropic and has numerous differentially expressed genes in vitro[35]. To assess whether the tcdR mutation is pleiotropic in 630Δerm, we performed RNAseq on wild type and the tcdR mutant grown for 18 h in TY media as an in vitro control. We found that of the 3548 protein-coding genes in the C. difficile 630Δerm genome, other than the genes of the Pathogenicity Locus (PaLoc), only two genes were differentially expressed (log2 fold change ±1 and adjusted p-value < 0.05) between the strains in vitro (CD1917, encoding EutE, and CD3087, encoding a transcription factor of the RpiR family) (Supplementary Data 1). When comparing in vivo expression profiles between the two strains, the majority of differentially expressed genes were detected at day 2, with 86 transcripts increased and 82 decreased in wild type relative to tcdR (Supplementary Data 1). After 4 days, wild type had 15 transcripts increased and 4 decreased. Consistent with the cecal content toxin activity assay, among the most significantly increased transcripts in wild type relative to the tcdR mutant were of the PaLoc. Interestingly, time was a more important variable with respect to the number of differentially expressed genes in vivo in both wild type and the tcdR mutant. When comparing wild type at day 4 to wild type at day 2, we found 249 genes had increased transcript levels and 155 decreased; the tcdR mutant at day 4 relative to day 2 had even more dramatic changes in gene expression, with 380 transcripts increased and 338 decreased (Supplementary Data 1). Together, these data show that inflammation is an important environmental determinant of the C. difficile transcriptome in vivo.

### Toxin-induced inflammation alters C. difficile metabolism.
We used prokaryotic gene set enrichment analysis to summarize the

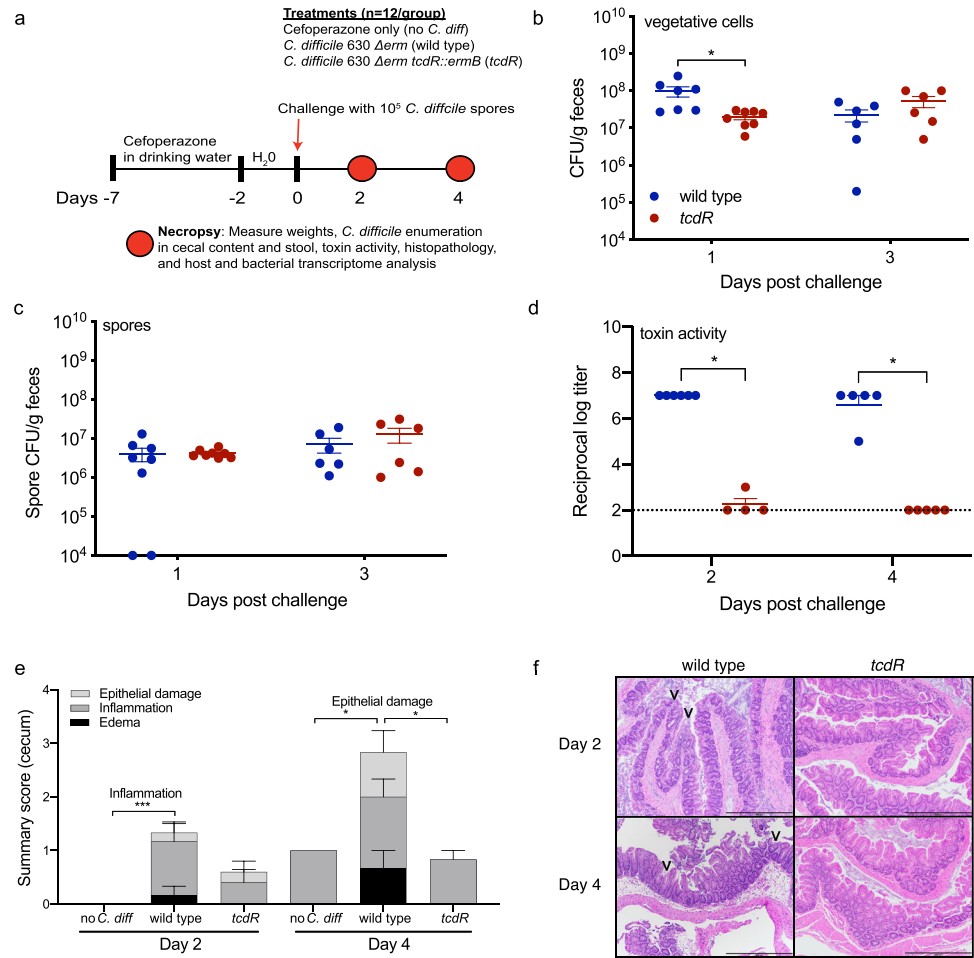

**Fig. 1 Inflammation is attenuated in *tcdR* mice in a mouse model of *C. difficile* infection. a** Schematic depicting experimental design. All mice ($n = 36$) received the antibiotic cefoperazone in their drinking water. Subsets of mice were orally gavaged with *C. difficile* 630Δ*erm* (wild type, $n = 12$) or *C. difficile* 630Δ*erm tcdR::ermB* (*tcdR*, $n = 12$) via oral gavage after antibiotic treatment. A group of mice were only treated with the antibiotic (no *C. diff* or uninfected, $n = 12$). **b** *C. difficile* vegetative cell CFUs in feces (wild type $n = 7$ on day 2 and $n = 6$ on day 4; *tcdR* $n = 8$ on day 2 and $n = 6$ on day 4, *$p = 0.0314$). **c** *C. difficile* spore CFUs in the feces (wild type $n = 8$ on day 2 and $n = 6$ on day 4; *tcdR* $n = 8$ on day 2 and $n = 6$ on day 4). **d** Toxin activity in the cecal content of mice (wild type $n = 6$ on day 2 and $n = 5$ on day 4; *tcdR* $n = 4$ on day 2 and $n = 5$ on day 4). For wild type vs. *tcdR* on day 2, $p = 0.0248$. For wild type vs. *tcdR* on day 4, $p = 0.02$. **e** Histopathological summary scores of the cecum ($n = 6$ mice per group per day). **f** Representative images of H&E stained ceca from mice in e; scale bar, 500 μm. Arrow heads indicate epithelial damage. The H&E staining was performed on each mouse ceca. All data in (**b–e**) are presented as the mean and error bars indicate SEM. Kruskal–Wallis test with Dunn's correction for multiple comparisons was used to test for statistical significance in (**b**), (**c**), and (**d**). Geissner-Greenhouse corrected ordinary Two-Way ANOVA with Tukey's multiple comparisons test was used in (**e**).

main patterns in differential gene expression (Fig. 2a, Supplementary Fig. 2, and Supplementary Data 2). Gene Ontology (GO) terms related to carbohydrate metabolism were enriched in wild type relative to *tcdR* at both days post-challenge, suggesting that wild type *C. difficile* may have had access to different carbohydrate nutrient sources. An operon encoding a phosphoenolpyruvate:carbohydrate phosphotransferase (PTS) system annotated to be specific to mannose/fructose/sorbose was significantly increased in wild type *C. difficile* at day 2 (Fig. 2b). PTS systems are typically induced by the presence of the carbohydrate that they import (and the absence of a repressing carbohydrate) via a transcriptional antitermination mechanism[37]. Expression of this operon normalized between wild type and the *tcdR* mutant by day 4, though at this timepoint the wild type had increased expression of genes predicted to be involved in fructose/mannitol and tagatose metabolism, as well as genes involved in extracellular polysaccharide production (Fig. 2b).

Among the most abundant GO terms in the set of transcripts that were decreased in wild type were those for oxidation-reduction and catalytic processes, as well as those for leucine biosynthesis.

The transcript levels for *ilvD*, involved in isoleucine and valine biosynthesis, were also decreased in wild type relative to *tcdR* (Fig. 2b). Both the *leu* operon and *ilvD* in *C. difficile* are transcriptionally repressed by CodY, whose repressive activity is high when bound by branched-chain amino acids and/or GTP[38,39]. Similar patterns of decreased expression in wild type were seen in the CodY-regulated operon encoding the machinery for the metabolism of succinate to butyrate (Fig. 2b). CodY has also been shown to positively regulate the expression of some genes in vitro, including *pflB*[39]. We found that *pflB* expression in vivo was increased at day 2 in wild type *C. difficile* (Fig. 2b). These data, when combined with previous metabolomic studies, suggest that the metabolomic environment of the inflamed gut may be enriched for specific carbohydrates and oxidative Stickland reaction substrates in the form of branched-chain amino acids.

Wild type *C. difficile* also had increased transcript levels of a number of genes involved in amino acid acquisition and metabolism compared to *tcdR* in vivo. One such gene, *CD3442*, encodes a putative M24 family Xaa-Pro prolidase. Eukaryotic prolidases are intimately linked to collagen metabolism, while

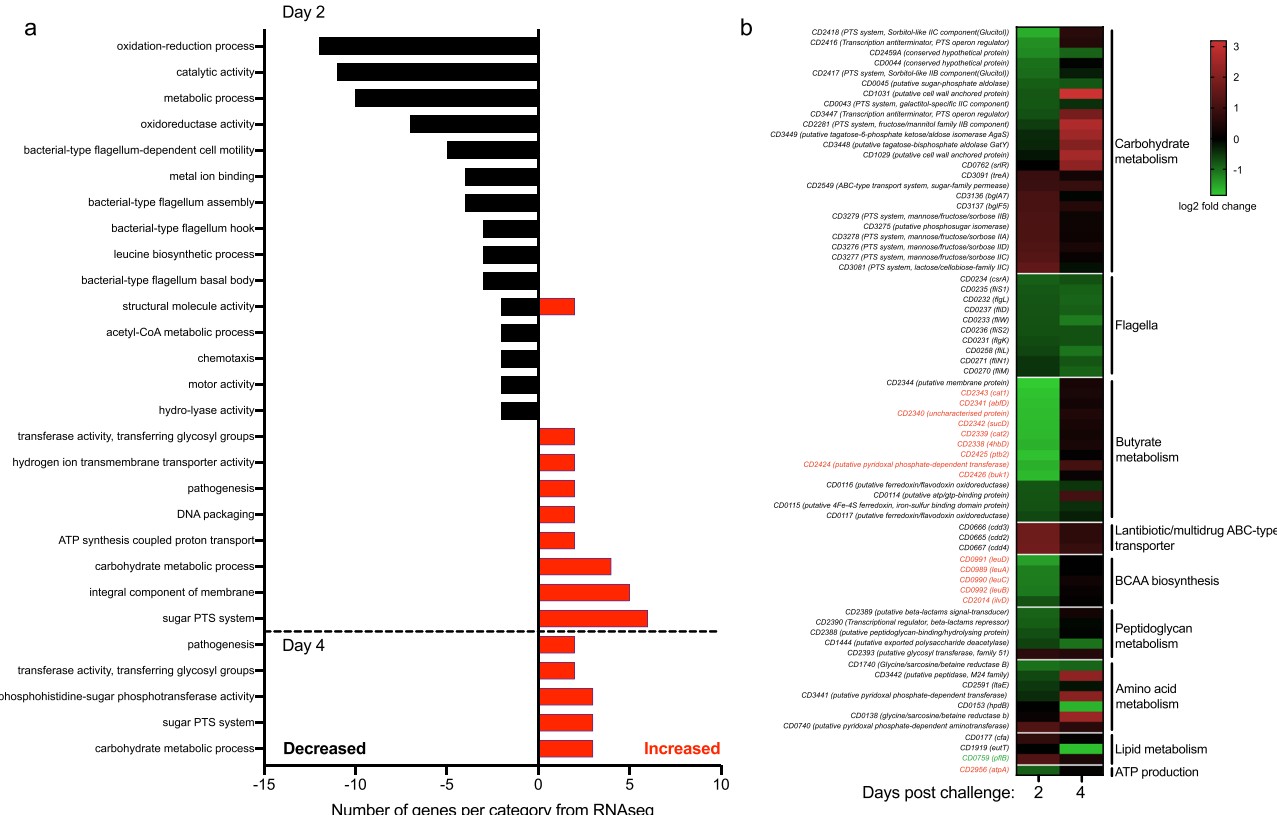

**Fig. 2 Metabolic gene expression in *C. difficile* is significantly altered by toxin-mediated inflammation. a** Gene set enrichment analysis of the differentially expressed genes in vivo from wild type *C. difficile* relative to the *tcdR* mutant from both days 2 and 4. GO terms that had transcripts with decreased levels are depicted in black bars and GO terms containing transcripts with increased levels are shown as red bars. **b** Heatmap of the log2 fold change of key operons and transcripts that were called as differentially expressed by DESeq2 (log2 fold change ±1 and adjusted $p < 0.05$) in wild type *C. difficile* ($n = 5$ on day 2, $n = 3$ on day 4) relative to *tcdR* ($n = 6$ on day 2, $n = 3$ on day 4). The labels of known CodY-regulated transcripts (according to Dineen, et al. J Bacteriology 2010) are color-coded in red if they increased in expression in a *codY* mutant in vitro and green if they decreased[39].

prokaryotic prolidases are often involved in protein turnover and proline recycling[40,41]. In contrast, the gene encoding 4-hydroxyphenylacetate decarboxylase (*hpdB*) was decreased in wild type at day 4; the transcript from the *hpdC* gene immediately downstream was also decreased ($p = 0.08$) (Fig. 2b). HpdB is involved in the fermentation of tyrosine to p-cresol, which has been shown to affect fitness in vivo in a murine relapse model and to modulate gut microbial community structure; the decreased *hpdB* transcript levels we observed may be consistent with lower levels of its substrate in the ceca of wild type mice[42–44]. A subset of genes identified as differentially expressed between wild type and the *tcdR* mutant were selected for qRT-PCR validation, which confirmed trends in expression from the RNAseq (Supplementary Fig. 3a–d).

**Gene expression for multiple aspects of *C. difficile* physiology is altered in the presence of inflammation**. Multiple genes encoding structural components of the flagella, including some that have been shown to induce inflammatory responses from host cells in vitro and in vivo, were decreased in expression in wild type (Fig. 2b). In contrast to flagellar genes, wild type had increased expression of the *cdd* operon, comprised of three genes (*cdd4*, *cdd3*, and *cdd2*) that are divergently transcribed from a two-component system (TCS) response regulator and histidine kinase (Fig. 2b). The *cdd* genes are annotated to encode the components of a multidrug/antibiotic ABC transport system, and given their genomic association with a TCS, they may encode an undescribed defense mechanism against antimicrobial peptides

that could function semi-analogously to the CprK-CprR/CprABC system previously described for *C. difficile*[45,46].

**Wild type *C. difficile* induces a robust inflammatory and proteolytic gene expression profile in host gut tissue.** To determine how the host responds to *C. difficile* toxin-induced inflammation, we compared the cecal tissue gene expression between three groups of mice (no *C. diff* or uninfected, wild type, and *tcdR*) with the NanoString Mouse Immunology panel modified to include probes targeting transcripts encoding matrix metalloproteinases (MMPs) and tissue inhibitors of metalloproteinases (TIMPs) (Fig. 3, Supplementary Figs. 4–6, and Supplementary Data 3). The top 50 differentially expressed transcripts (in terms of significance) from the wild type relative to *tcdR* comparisons from both days were combined and plotted in a heatmap with hierarchical clustering of samples, and two distinct clusters were observed (Fig. 3a, Supplementary Figs. 7–8). All samples from *tcdR* mice and uninfected controls formed one large cluster, while all wild type mice formed their own distinct cluster; neither cluster showed sub-clustering based on time points. Only 42% of the differentially expressed genes in wild type mice relative to *tcdR* mice were significant at both days 2 and 4, suggesting that the nature of the immune response to wild type *C. difficile* changed over the course of infection (Supplementary Fig. 5). In contrast to wild type mice, *tcdR* mice had no significant differentially expressed genes in their cecal tissue when compared to uninfected controls (Supplementary Fig. 4g–h). These data are consistent with the histopathological analysis, and show that in

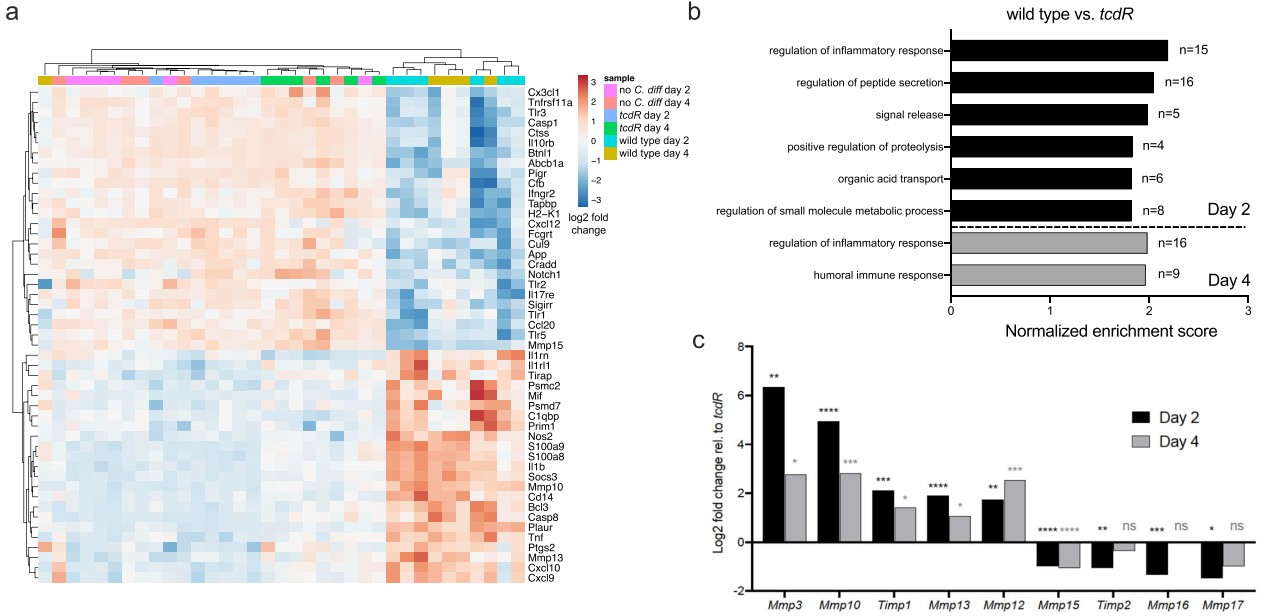

**Fig. 3 C. difficile induces expression of numerous transcripts associated with inflammation and ECM degradation. a** Heatmap of the top 50 differentially regulated transcripts (by adj. *p*-value) in the ceca of uninfected control (no *C. diff*), wild type, and *tcdR* mice (*n* = 5–6 mice per group per day). **b** Gene set enrichment analysis of the differentially expressed genes in wild type mice relative to *tcdR* mice. **c** Log2 fold changes of various *Mmp*s and associated transcripts from wild type vs. *tcdR* mouse ceca. Significance for all transcripts except *Mmp3* was determined using differential expression analysis within the NanoString nSolver Advanced analysis software (log2 fold change ±1 and adjusted *p* < 0.05). *Mmp3* expression levels were determined via qRT-PCR on cDNA generated from the same RNA used in the NanoString analysis, and individual data points can be seen in Supplementary Fig. 6c (Kruskal–Wallis with Dunn's correction for multiple comparisons). **p* < 0.05, ***p* < 0.01, ****p* < 0.001, *****p* < 0.0001.

the absence of toxin activity, the *tcdR* mutant is relatively inert in vivo with respect to stimulating a host immune response.

We next performed gene set enrichment analysis for each group of differentially expressed genes, using GO biological process terms and considering the direction of expression for each transcript (Fig. 3b and Supplementary Data 4). The GO term for regulation of inflammatory processes was enriched in transcripts with increased abundance in cecal tissue from wild type mice at both days relative to *tcdR* mice. The second most enriched at day 2 was regulation of peptide secretion, consistent with the role of host-derived antimicrobial peptides being produced as an arm of the innate immune response. Of particular interest was the enrichment of genes involved in the positive regulation of proteolysis, as peptide fragments derived from these processes may serve as nutrient sources for *C. difficile*, which is well-known for using amino acid fermentation as an energy source in vitro and in vivo[11,12,36,47–51]. In addition to the upstream regulators of various proteolytic processes, cecal tissue from wild type mice had significant increases in transcripts from genes encoding multiple MMPs including *Mmp3*, *Mmp10*, *Mmp12*, and *Mmp13*; probes targeting *Mmp3* were not included in the NanoString custom panel, so fold change for this transcript is reported based on qRT-PCR results (Fig. 3c, Supplementary Fig. 6c). A number of transcripts were selected for further analysis via qRT-PCR, which confirmed expression patterns observed via the NanoString approach (Supplementary Fig. 6a–e). Taken together, these data show that *C. difficile* toxin activity induces a highly inflammatory gut environment, and implicate MMP substrates, such as collagen and other ECM components, as reservoirs of Stickland substrate amino acids in vivo.

**Toxin activity induces collagen degradation which supports *C. difficile* growth in vitro.** The increase in numerous *Mmp* transcripts during toxin-induced inflammation suggested the ECM

may be altered during CDI. We chose to examine whether toxin activity on cells in vitro affected collagen integrity, as it is highly abundant and an excellent source of proline, hydroxyproline, glycine, and alanine, all of which are amino acids that *C. difficile* can ferment via the Stickland reaction[14]. Confluent ECM-producing IMR90 human fibroblast monolayers were cultured on 24-well plates for 3 days prior to toxin treatment. Collagen remodeling was then visualized using immunofluorescence (Fig. 4a). By confocal microscopy, we observed that the toxins caused a notable disruption of the collagen network over a 12 h period. Extensive networks of collagen fibrils were apparent in untreated cells, whereas collagen in toxin-treated cells appeared fragmented and condensed into globular structures. Moreover, we observed a significant decrease in collagen fluorescence over a 15 h period, validating that collagen was being degraded in the presence of toxins (Mann–Whitney test *p* < 0.0001) (Fig. 4b).

Given that toxins induced the degradation of collagen in IMR90 cells, we next speculated that *C. difficile* can acquire nutrients from degraded collagen. To test this, *C. difficile* was grown in a minimal media with proline, without proline, or without proline and supplemented with heat-degraded collagen for 24 h (Fig. 4c). Although not statistically significant (Kruskal–Wallis with Dunn's correction for multiple comparisons), *C. difficile* grew approximately ten-fold higher in media supplemented with degraded collagen compared to media with no proline, indicating collagen can provide a source of proline for *C. difficile* growth. Additionally, when Pro-Gly and Gly-Pro dipeptides (abundant in host collagen) were substituted for collagen degradation products, *C. difficile* grew to levels comparable to the standard minimal media control after 24 h (*p* = 0.0015 for both conditions compared to their respective 0 h timepoint CFUs, two-way ANOVA with Sidak's multiple comparisons test) (Fig. 4d). This suggests that *C. difficile* can exploit host collagen degrading activity as a means to acquire peptides and amino acids.

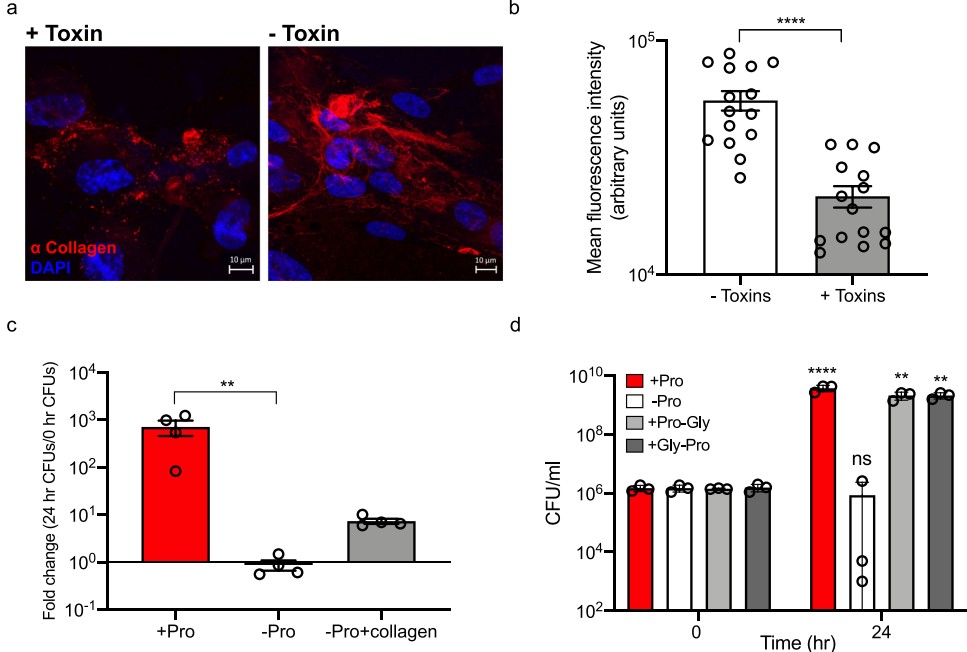

**Fig. 4 Toxin-mediated degradation of collagen supports *C. difficile* growth in vitro. a** Representative images of DAPI (blue) and collagen (red) produced by IMR90 cells. Confluent cell monolayers were treated with 0.5 pM TcdA and TcdB and images were collected 12 h later. Collagen was stained with a mix of antibodies against collagen types I, III, and V in a 1:1:1 ratio; scale bar, 10 μm. **b** Mean fluorescent intensity of Alexa Fluor 568 stained collagen produced by IMR90 cells cultured in the presence or absence of 0.5 pM TcdA and TcdB for 15 h calculated using ImageJ software ($n = 15$ fields of view for each condition). Statistical significance was determined by Mann–Whitney rank-sum test; ****$p < 0.0001$. **c** *C. difficile* was grown in complete CDMM ($n = 4$), CDMM lacking proline ($n = 4$), or CDMM lacking proline and supplemented with heat-degraded collagen ($n = 4$). CFUs/ml were enumerated at 0 and 24 h; **$p = 0.0051$. Kruskal–Wallis test with Dunn's correction for multiple comparisons was used to test for statistical significance. **d** *C. difficile* was grown in complete CDMM ($n = 4$), CDMM lacking proline ($n = 4$), or CDMM lacking proline supplemented with purified Pro-Gly ($n = 4$) or Gly-Pro ($n = 4$) dipeptides. 24 h CFUs were compared to the 0 h CFUs for statistical tests; **$p = 0.0015$, ****$p < 0.0001$. All data in (**b**–**d**) are presented as the mean and error bars indicate SEM. Statistical significance was determined by two-way ANOVA with Sidak's multiple comparisons test.

**Toxin-mediated inflammation suppresses the return of the Bacteroidaceae in the gut microbiota.** Given the importance of the microbiota in rendering the gut an inhospitable environment for *C. difficile*, we hypothesized that toxin-mediated inflammation may exclude or suppress members of the gut microbiota that contribute to colonization resistance, and/or select for microbes that may benefit *C. difficile* through further niche modification/ preservation, or other mechanisms like cross-feeding. We performed 16S rRNA amplicon sequencing on cecal DNA from mice challenged with wild type, the *tcdR* mutant, and uninfected controls (Supplementary Data 5 for amplicon sequence variant (ASV) relative abundances and taxonomy). As expected, a significant driver of community similarity was *C. difficile* colonization status; however, a number of mice from the uninfected controls and the *tcdR* mice had community structures with an increased abundance of ASVs from the Akkermansaceae (at day 2) and Bacteroidaceae (both days) Families, which were low or undetected in the cecal microbiota of wild type mice (Fig. 5a). The Bacteroidaceae were detected in samples from the same cages over time, suggesting a potential cage effect. Additionally, a number of low abundance Family members, including the Coriobacterales, Paenibacillaceae, and Burkholderiaceae, were present in either uninfected controls, *tcdR* mice, or both, but undetected in the cecal microbiota of wild type mice (Supplementary Data 5). ASVs from the Staphylococcaceae Family were detected at relatively high abundance in uninfected controls at both days, but were in low abundance or undetected in mice challenged with either strain of *C. difficile* (Fig. 5a). ASVs generated by DADA2 were clustered into operational taxonomic units (OTUs) at 99% sequence identity, and Hellinger-transformed OTU abundances

were analyzed by PCA to determine the similarity of each cecal community (Fig. 5b). At day 2, both wild type mice and all but one of the uninfected controls formed tight, distinct clusters, while *tcdR* mice showed no specific clustering. By day 4, no group clustered very closely. Some wild type and *tcdR* mice community structures were driven by ASVs from various Lachnospiraceae and Erysipelotrichaceae Families, while community structures of other *tcdR* and uninfected mice were driven by ASVs from the Bacteroidaceae, Akkermansaceae, and Staphylococcaceae Families. These data show that colonization with *C. difficile* and toxin-mediated inflammation can significantly impact the return of the gut microbial community structure.

The ASV from the Bacteroidaceae Family identified in this study was classified at the genus level as *Bacteroides*. Since the Bacteroidaceae were more abundant in some mice in the absence of inflammation, and hydroxyproline utilization genes are enriched in this Family, we sought to characterize the growth of two representative members, *Bacteroides thetaiotaomicron* and *Bacteroides fragilis*, in a minimal media with and without supplementation of substrates enriched in collagen, proline or hydroxyproline (Fig. 5c–d)[52]. Supplementation of the minimal media with either amino acid led to approximately ten-fold increased growth for both species over 16 h relative to unsupplemented media. Glucose was required for robust growth, however when minimal media with glucose was supplemented with either amino acid there was almost two-fold higher growth of *B. thetaiotaomicron* and *B. fragilis* relative to glucose alone ($p < 0.001$ and $p < 0.0001$, One-way ANOVA with Tukey's multiple comparisons test). This trend of increased growth in supplemented media indicates that both proline and hydroxyproline can

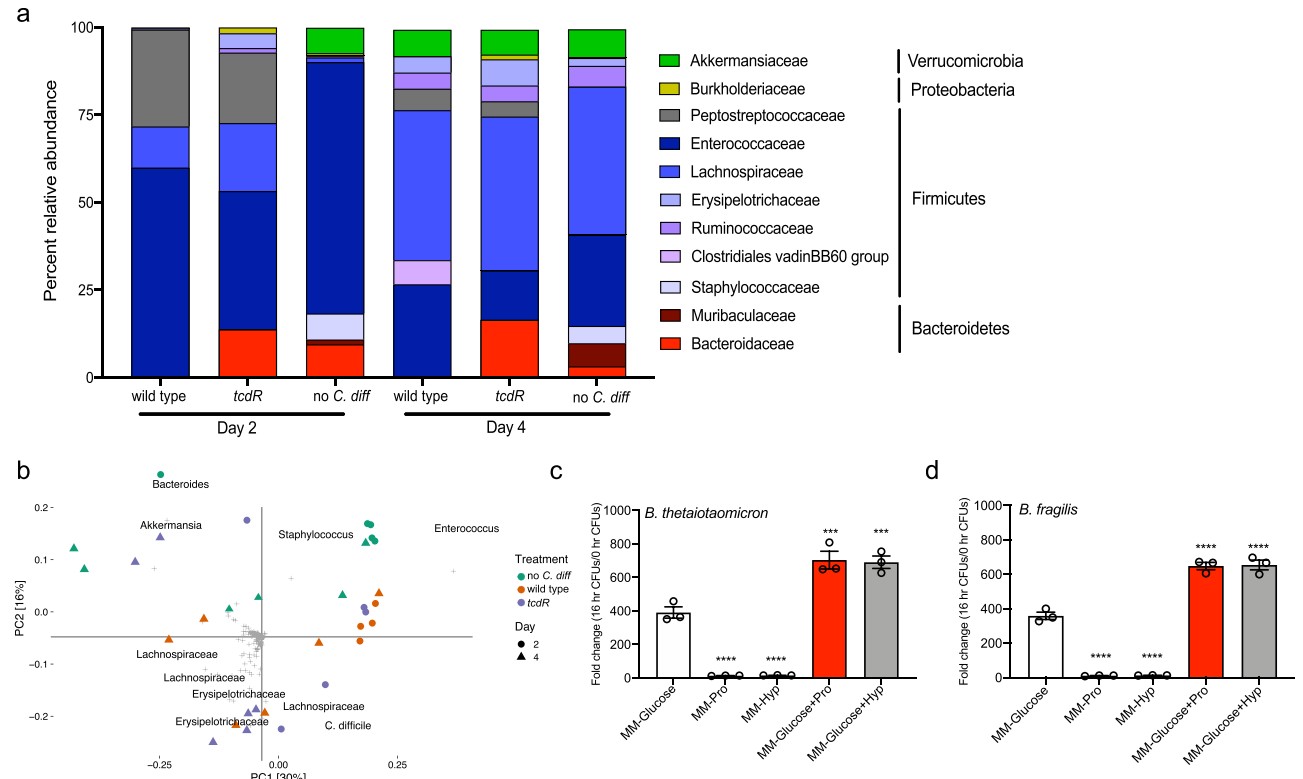

**Fig. 5 C. difficile toxin activity suppresses the Bacteroidaceae that are able to compete with C. difficile for amino acids. a** Averaged percent relative abundance of Family level ASVs in each treatment group per timepoint. ASVs with less than 1% relative abundance in all samples were not included. **b** PCA biplot of 16S rRNA amplicon sequences derived from cecal tissue from uninfected or no C. diff control mice ($n = 5$ on day 2, $n = 6$ on day 4), wild type mice ($n = 4$ on day 2, $n = 6$ on day 4) and tcdR mice ($n = 5$ on day 2, $n = 6$ on day 4). Each colored symbol represents an individual mouse's cecal microbiome, with circles being those from day 2 and triangles from day 4. 99% of OTUs are shown as gray crosses; the 10 OTUs furthest from the origin are labeled by the finest taxonomic rank identified (family, genus, or species). **c** 16 h fold change in CFUs of B. thetaiotaomicron in minimal media with ($n = 3$) or without glucose ($n = 3$), supplemented with either proline ($n = 3$) or hydroxyproline ($n = 3$) ****$p = 0.0003$, MM-Glucose vs. MM-Glucose+Pro; MM-Glucose vs. MM-Glucose+Pro, ***$p = 0.0005$; ****$p < 0.0001$. **d** 16 h fold change in CFUs of B. fragilis in identical media conditions as in (**c**) ($n = 3$ for each condition) ****$p < 0.0001$. All data in (**c**) and (**d**) are presented as the mean and error bars indicate the SEM. Statistical significance in (**c**) and (**d**) was determined by One-way ANOVA with Tukey's multiple comparisons test.

be utilized by members of the genus *Bacteroides*, and that these amino acids may be valuable nutrients to compete for in the gut.

**Toxin-mediated alterations of host *Mmp* expression and the gut microbiota are conserved in mice challenged with epidemic *C. difficile* R20291 strain.** As the wild type strain used in this study (*C. difficile* 630Δ*erm*) is a multi-passaged, erythromycin sensitive lab strain, we sought to replicate our findings in the clinically relevant *C. difficile* R20291 strain. A Δ*tcdR* mutant was constructed via allelic replacement; it and the parent strain were used to challenge antibiotic treated mice (Fig. 6). No differences in total fecal *C. difficile* load (vegetative cells + spores) were observed between the strains at any day post challenge, however, significantly fewer Δ*tcdR* spores were recovered from feces at day 2 ($p < 0.0001$, Kruskal–Wallis with Dunn's correction for multiple comparisons), consistent with a previous report on a pleiotropic *tcdR* ClosTron mutant in R20291 showing decreased sporulation efficiency in vitro (Fig. 6a–b)[35]. Similar to what was observed for *C. difficile* 630Δ*erm* and its *tcdR::ermB* derivative, cecal content from mice challenged with wild type R20291 had significantly higher toxin titers than that from Δ*tcdR* at both days (Fig. 6c). Importantly, comparable patterns of increased expression of the same *Mmp*s were observed in cecal tissue from wild type R20291 mice relative to the Δ*tcdR* mice (Fig. 6d). Given the pleiotropic nature of the *tcdR* mutation in the R20291 strain, we elected not to

perform *C. difficile* gene expression studies in vivo. 16S rRNA amplicon sequencing on cecal tissue isolated at day 4 showed similar patterns in the microbial community structures as observed in mice challenged with the 630Δ*erm* strains. Wild type R20291 mice had very low or undetectable Bacteroidaceae ASVs, while Δ*tcdR* mice had considerably higher levels (2% average relative abundance vs. 32%, respectively) (Fig. 6e and Supplementary Data 6). The ASV designated as *Bacteroides* in this study is identical to the *Bacteroides* ASV that was identified in the cecal communities of uninfected controls and *tcdR* mice from the 630Δ*erm* study in Fig. 5a–b. Collectively, these data show that induction of host *Mmp* gene expression is a conserved component of the immune response to *C. difficile* toxin activity, and that the decreased levels of Bacteroidaceae in the inflamed gut may be biologically significant and an additional mechanism where *C. difficile* is able to exploit a niche and thrive due to host inflammation.

## Discussion
*C. difficile* is a major nosocomial pathogen and cases of CDI are now being diagnosed in individuals who lack the classic predisposing traits of recent antibiotic use or compromised immune status[53]. While effective treatments exist for *C. difficile*, some patients require fecal microbiota transplants (FMTs) to resolve their infections, highlighting the need to better understand how this pathogen creates a niche for persistence in a host. CDI is

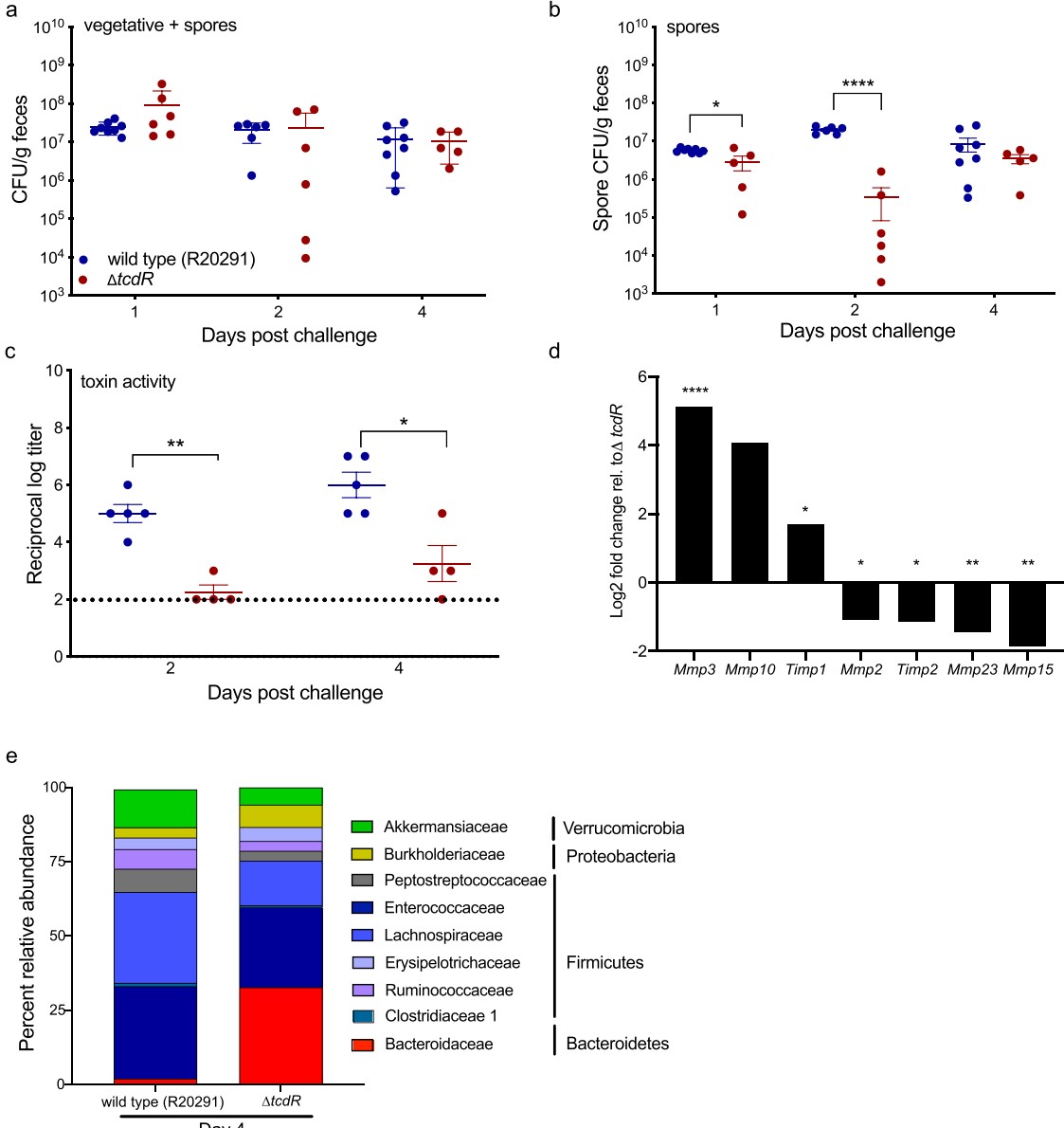

**Fig. 6 C. difficile R20291 toxin activity similarly shapes the host gut transcriptome and microbiota community structure in mice. a** Total *C. difficile* CFUs (vegetative and spores) in feces over time ($n = 8$ mice for WT R20291 and $n = 6$ for or Δ*tcdR* mice; note that not every mouse provided a stool sample). **b** Fecal spore CFUs over time ($n = 6$–8 mice per group per day, *$p = 0.0318$). **c** Toxin activity in the cecal content of R20291 or Δ*tcdR* mice, as assessed by the Vero cell cytotoxicity assay ($n = 5$ mice per group on day 2, $n = 4$ mice per group on day 4, *$p = 0.0383$, **$p = 0.0055$). **d** Log2 fold change of *Mmp* and *Timp* expression ($n = 3$ mice per group) derived from NanoString transcriptome analysis (log2 fold change ±1 and adjusted $p < 0.05$). **e** Average percent relative abundances of 16S rRNA amplicon sequences from cecal tissue isolated at day 4 ($n = 5$ mice per group). One-way Kruskal–Wallis test with Dunn's correction for multiple comparisons was used to test for statistical significance for (**a**) and (**b**). A mixed effects model with the Geissner-Greenhouse correction and Sidak's multiple comparisons test was used for (**c**). Significance was determined in (**d**) by NanoString nSolver Advanced analysis software.

highly inflammatory, therefore we leveraged the power of bacterial genetics with *tcdR* mutants in two strains of *C. difficile* to define how toxin-mediated inflammation alters the gut environment in vivo, with a focus on bacterial metabolism and the gut microbiome. While the host and microbiota responses are largely consistent between the strains, we observed that *C. difficile* R20291 Δ*tcdR* did not have a CFU defect in vivo. The R20291 genome encodes the *cdt* binary toxin locus, which is present and presumably functional in both the wild type and the Δ*tcdR* mutant; inflammation induced by the binary toxins may alter the nutrient pool to support increased growth of Δ*tcdR*[54,55].

We observed changes in numerous *C. difficile* metabolic genes in vivo, consistent with the hypothesis that toxin-induced inflammation alters the nutrient landscape in the host gut environment. In particular, we found time-dependent increases and decreases in the expression of multiple PTS carbohydrate import genes, and decreased expression of CodY-regulated genes for branched-chain amino acid biosynthesis and butyrate production in wild type *C. difficile*. On its face, this represents a paradox: CodY de-repression of *tcdR*, *tcdA*, and *tcdB* leads to extremely high levels of the toxins, yet in our study we found that a number of CodY-regulated genes were repressed in vivo in the

presence of inflammation. Two possible mechanisms may explain this discrepancy: phase variation in the expression of *sigD*, a positive regulator of *tcdR*, and bimodal expression of *tcdR* and the toxin genes[34,56]. This indicates that a larger fraction of the wild-type *C. difficile* population may have increased access to nutrients that sustain growth and thus would have increased CodY repressive activity. Future studies examining the per-cell expression levels of the CodY-regulated genes identified in our RNAseq studies are warranted. Additionally, we have identified numerous targets for further study via mutagenesis with respect to *C. difficile* metabolic requirements in vivo, or to understand how it resists the deleterious effects of host inflammation (*cdd* operon). Further, our data highlight the need to fully understand the spectrum of behavior of individual bacterial cells across a population during the infection process. This may open novel avenues for therapeutic targeting of specific subsets of pathogens within a metabolically heterogeneous population.

Our data support a model where the activity of the toxins stimulate an inflammatory host response that includes gene expression signatures consistent with degradation of collagen and other components of the ECM. Collagen is rich in Stickland reaction substrates like proline (and hydroxyproline, which *C. difficile* can dehydrate to proline) and glycine, amongst other amino acids that *C. difficile* can ferment[13,52,57]. Hence, the ECM and collagen may serve as a reservoir of preferred amino acid nutrients that sustain the metabolic burden of large bacterial populations and the production of the toxins over time within the host gut environment. Purified TcdB has been reported to stimulate MMP2 activity in bovine smooth muscle cells in vitro, suggesting that increased expression and activity of MMPs may be a general consequence of disruption of the actin cytoskeleton by *C. difficile* toxin activity[58]. Aberrant MMP activity has been reported as a factor in the pathogenesis of IBD, and IBD patients may be more likely to contract CDI than the non-IBD population, suggesting that ECM remodeling could contribute to creating a niche in humans that *C. difficile* can more readily colonize and thrive[6–10]. Additionally, the presence of prior inflammation is sufficient to render mice more susceptible to colonization by *C. difficile* and results in more severe disease[59,60]. The gastrointestinal pathogens *S. enterica* and *V. cholerae* have also been shown to benefit from inflamed host tissue, gaining access to nutrients that increase their fitness as pathogens[3–5,61,62]. Future studies are needed to determine the extent to which MMP activity contributes to the peptides and amino acids that *C. difficile* has access to in the inflamed gut.

Lastly, we show that the community structure of the gut microbiota is altered by the presence of *C. difficile* and the activity of its toxins, supporting the hypothesis that inflammation can benefit *C. difficile* by selecting against competitors and for potential allies. In particular, we found that the Bacteroidaceae tended to bloom in the ceca of uninfected controls, or mice challenged with two *tcdR* mutants from phylogenetically divergent *C. difficile* strains[63]. In support of this, negative associations between *C. difficile* and members of the Bacteroidaceae have been reported in human studies, as well as in vitro, and in mouse models of CDI[64–68]. Further work is necessary to identify which mediator(s) of host inflammation are responsible for the restriction of the Bacteroidaceae in our model of CDI. *B. thetaiotaomicron* has been shown to cross-feed *C. difficile* with succinate in a co-colonization model and our RNAseq studies support this, in that *tcdR* mice colonized with the Bacteroidaceae had increased expression of the succinate to butyrate operon[69].

Homologs of the *C. difficile* hydroxyproline dehydratase gene, *hypD*, are enriched in the Bacteroidaceae, and *C. difficile* can satisfy its proline requirements through the utilization of hydroxyproline in vitro[52]. Given the high levels of hydroxyproline

in collagen, it is possible that toxin-mediated exclusion or suppression of the Bacteroidaceae removes a competitor for a vital nutrient. In support of their role as potential competitors, a recent report found that a five-member cocktail of mucin saccharide metabolizing bacteria that included *Akkermansia* and *Bacteroides* was able to reduce the *C. difficile* burden in a murine model of experimental CDI[69]. While we focused on proline and hydroxyproline in this study, our RNAseq data suggests that a number of other amino acids and carbohydrates may be more abundant in the *C. difficile* toxin-inflamed gut. Recent studies have highlighted the role of members of the microbiota in providing colonization resistance to enteric pathogens via competition for both carbohydrates and amino acids[69–73]. Taken together, our data suggest that *C. difficile* may preemptively compete by remodeling the gut environment to supply nutrients to itself and to prevent the return of competitors via an intense inflammatory environment.

Use of a *tcdR* mutant combined with omic technologies and in vivo models represents a powerful approach for asking how *C. difficile* toxin-induced inflammation alters the host gastrointestinal environment in ways that may create or preserve a niche during colonization and disease. These results provide multiple avenues for future study of the basic biology of CDI at the level of host response, pathogen response to inflammation, and manipulation of the host gut microbiota. While it was outside the scope of this work, we think the approach of querying the gut microbiota of mice colonized with wild type and mutant strains of *C. difficile*, in particular mutants in key metabolic pathways, may be fruitful for identifying bacterial taxa that bloom in the presence of a nutrient(s) that a mutant *C. difficile* population can no longer use. This approach may, with enough mutants in important metabolic pathways, contribute to a rationally designed consortium of bacteria that could compete with *C. difficile* for essential nutrients in models of colonization and disease.

## Methods

**Bacterial strains, growth conditions, and mutagenesis**. *C. difficile* strains 630Δ*erm* and an isogenic *tcdR*::*ermB* ClosTron insertion mutant (both kindly gifted by Rita Tamayo), as well as the R20291 strain and its isogenic Δ*tcdR* mutant were routinely grown in and on Brain Heart Infusion (BHI) or Tryptone Yeast (TY) broth and agar; plates and cultures were grown at 37 °C in an anaerobic chamber (Coy). For genetic manipulation of *C. difficile*, strains were grown on and in BHI agar and broth supplemented as necessary with 10 μg/ml of thiamphenicol, 50 μg/ml of kanamycin, and 16 μg/ml of cefoxitin to select for transconjugants or thiamphenicol alone for plasmid maintenance. Samples derived from in vivo studies were plated onto CCFA (cefoxitin, cycloserine, fructose agar) to select for and enumerate vegetative *C. difficile* CFUs, and TCCFA, containing the germinant taurocholate to enumerate spore CFUs.

The R20291 Δ*pyrE* strain was used to construct the Δ*tcdR* mutant[16]. Briefly, ~1.2 kb upstream and downstream of the *tcdR* gene were PCR amplified with Phusion High-Fidelity DNA polymerase (NEB M0530S). All primers used for the construction of the mutagenesis vector and for PCR screening of transconjugants can be found in Table 1. The homology arms were then combined into one single linear fragment via splice overlap extension PCR, then A-tailed with Taq polymerase (NEB M0267S). The A-tailed product was ligated into pCR2.1 (Thermo Fisher K202020); the resulting plasmid was digested with BamHI and KpnI and the ~2.4 kb homology arm fragment was ligated into the corresponding sites in pMTL_YN4 using T4 DNA Ligase (NEB M0202S). The final plasmid was conjugated into the R20291 Δ*pyrE* strain with *E. coli* SD46, and thiamphenicol resistant large colony variants were screened by PCR for plasmid integration into the chromosome. After confirming plasmid integration, colonies were grown in BHI broth overnight in the absence of selection to allow for plasmid excision, then plated onto minimal media agar supplemented with 5 μg/ml uracil and 2 mg/ml of 5-fluoroorotic acid to select for bacteria that had excised and lost the mutagenesis vector. Individual colonies were re-streaked twice on the same selective media, then PCR screened for loss of *tcdR*.

**Spore preparation**. Spores were prepared as in Edwards and McBride[17]. Five hundred microliters of mid-log phase cultures was spread onto 70:30 agar plates and incubated at 37 °C for 4 days, after which time the plates were removed from the anaerobic chamber. The bacterial lawns were scraped off and suspended in 10 mL sterile PBS, mixed 1:1 with 96% ethanol, vortexed vigorously for 30 s, and

**Table 1 Primers used in this study.**

| Primers 5′ to 3′ (lowercase for restriction sites) | Target |
|---|---|
| 5′tcdR.up.bamHI - ggatccTATATGAAAGAAGAGCATAATTTACCAG | *tcdR* upstream homology arm |
| 3′tcdR.up.SOE - TCATTAATTACATAAAATCATCCTCTCTTATATTTATAATG | |
| 5′tcdR.down.SOE - GGATGATTTTATGTAATTAATGAATTTAAAGAAATATTTACAATAG | *tcdR* downstream homology arm |
| 3′tcdR.down.kpnI - ggtaccATATACACCACCAACTTCTTTTAAGGC | |
| 5′tcdRc.bamh1 - ggatccGATTTCATAAAAGATACTATTTTAGTCTTG | Check for loss of *tcdR* coding sequence |
| 3′tcdRc.kpn1 - ggtaccGTTAATTCTAAAATTTGATTTCTATTG | |
| tcdR.left.up.gDNA - CAATGTTAGAAAATCATTTGAGTG | Check for knockout plasmid integration |
| YN4.kpn.side - ATGACCATGATTACGAATTCG | |
| YN4.bamHI.side - GCGTGACGTCGACTCTAG | Check for knockout plasmid integration |
| tcdR.right.down.gDNA - ATATCAAAATGCTCTGAAGTATATCC | |
| 5′mB.actin.intron1 - GCCTTCTTTTGTGTCTTGATAG | Check for host gDNA |
| 3′mB.actin.exon3 - CTGGGTCATCTTTTCACG | |
| 5′plaur.q - GGCACAGCAGGTTTCCATA | *Plaur* |
| 3′plaur.q CGGTGGAAAGCTCTGAAGAT | |
| 5′S100a9.q - CTCCTCAAAGCTCAGCTGATTG | *S100a9* |
| 3′S100a9.q - AATGGTGGAAGCACAGTTGG | |
| 5′c9.q - CCTGAAAGAGAAGATTCTCAGAGG | *C9* |
| 3′c9.q - CGTTTGCCAGGGACGAG | |
| 5′tlr5.q - CGCCTCATCTCACTGCATAC | *Tlr5* |
| 3′tlr5.q - ACAGATGTGTCTGGCATATGTT | |
| 5′mmp3.q - GGGATGATGATGCTGGTATG | *Mmp3* |
| 3′mmp3.q - TGACAATAAGACTACTGTCCTTT | |
| 5′mmp12.q - CTAGAAGCAACTGGGCAACT | *Mmp12* |
| 3′mmp12.q - GCTCTAAGATGCTGTACATCGG | |
| 5′mmp13.q - CTACCCACTTGTTCTAATGACCTAT | *Mmp13* |
| 3′mmp13.q - GCTGTGTCTTAGCTGGATCTAC | |
| 5′rpoC.qPCR - TGGCAGTCCATGTACCTTTATC | *rpoC* |
| 3′rpoC.qPCR - GGTGAACCATCTTTAGGAGCA | |
| 5′cdd3.q - ATCCAGATATGTTAGGTGGATTTGA | *cdd3* |
| 3′cdd3.q - AGCTTCCATGTATCTCCTTTATGT | |
| 5′PTS_fms.q - AGAATTGTCCAGAGAGCTTGTT | *PTS IIA* |
| 3′PTS_fms.q - TCTCAGCATCATCTGCTGATTTA | |
| 5′ilvD.q - CATGAACTCTTGTCCTGGATGT | *ilvD* |
| 3′ilvD.q - GTGACGCAGCAGTACCATTA | |
| 5′cat1.q TGGATTTACACCTTCAGGCTATC | *Cat1* |
| 3′cat1.q - CTCCATCAACTTCTGGACCTAAA | |
| 5′ CD630DERM_34420.q - GCCACATGGAAGACCTACAA | *CD630DERM_34420* |
| 3′ CD630DERM_34420.q - ACGAGTCATGTCTGATTGATACC | |
| 5′tcdA.qPCR - ACTAGACGAACATGACCCATTAC | *tcdA* |
| 3′tcdA.qPCR - GCTACCGTTGCAGCTATAGATAA | |

allowed to sit at room temperature on the benchtop for 1 h. The suspension was centrifuged at 3000 rpm for 10 min. The pellet was suspended in 10 ml fresh sterile PBS, and centrifuged again; this was repeated twice. The final pellet was suspended in 1 ml PBS and serial dilutions were plated on BHI agar with 0.1% of the germinant taurocholate for spore CFU enumeration. Spore stocks were also enumerated one day prior to the day of challenge to confirm spore stock CFUs prior to making the inocula for in vivo studies; inocula were also diluted and plated the day of challenge.

**Animals and housing.** C57BL/6J WT mice (5–8 weeks old; $n = 18$ male and $n = 18$ female) were purchased from Jackson Labs. The food, bedding, and water were autoclaved, and all cage changes were performed in a laminar flow hood. The mice were subjected to a 12 h light and 12 h dark cycle. Mice were housed in a room with a temperature of 70 F and 35% humidity. Animal experiments were conducted in the Laboratory Animal Facilities located on the NCSU CVM campus. Animal studies were approved by NC State's Institutional Animal Care and Use Committee (IACUC). The animal facilities are equipped with a full time animal care staff coordinated by the Laboratory Animal Resources (LAR) division at NCSU. The NCSU CVM is accredited by the Association for the Assessment and Accreditation of Laboratory Animal Care International (AAALAC). Trained animal handlers in the facility fed and assessed the status of animals several times per day. Those assessed as moribund were humanely euthanized by $CO_2$ asphyxiation.

**Mouse model of *C. difficile* infection.** The mice were given 0.5 mg/mL cefoperazone in their drinking water for 5 days to make them susceptible to *C. difficile* infection[18,19]. The mice were then given plain water for 2 days, after which time they ($n = 12$, males and females) received $10^5$ spores of either *C. difficile* 630Δerm

(wild type) or *C. difficile* 630Δerm *tcdR::ermB* (*tcdR*) via oral gavage (Fig. 1a). One group of mice ($n = 12$, males and females) received antibiotics and no *C. difficile* (no *C. diff* or uninfected) spores. Mice were weighed daily and monitored for clinical signs of distress (ruffled fur, hunched posture, slow ambulation, etc.). Fecal pellets were collected 1 and 3 days post-challenge and diluted 1:10 w/v in sterile PBS, then serially diluted in 96-well PCR plates and plated onto CCFA for vegetative *C. difficile* CFU enumeration. The serially diluted samples were then removed from the anaerobic chamber and heated to 65 °C for 20 min to kill vegetative cells; the heat-treated dilution plate was passed back into the anaerobic chamber and the dilutions were plated onto TCCFA to enumerate spore CFUs.

At days 2 and 4 post-challenge, mice were humanely sacrificed ($n = 6$ per treatment), and necropsy was performed. Cecal content was harvested for enumeration of vegetative *C. difficile* and spore CFUs, as well as for RNA extraction and toxin activity. Cecal tissue was harvested for RNA extraction for gene expression analysis, 16S rRNA sequencing, and histopathology. Colon tissue was also harvested for histopathology. Samples for sequencing and toxin activity were immediately flash frozen in liquid nitrogen and stored at −80 °C until processing. Toxin activity in the cecal content was quantified using the Vero Cell cytotoxicity assay[19]. Briefly, the content was diluted 1:10 w/v in sterile PBS, and 10-fold dilutions were added to Vero cells in a 96-well dish for ~16 h. The reciprocal of the lowest dilution in which ~80% of the cells have rounded was reported as the titer.

The R20291 study was conducted similarly to the one described above with some minor differences. C57BL/6J mice (5–8 weeks old, $n = 14$ male and 14 female) were orally gavaged with $10^5$ spores of *C. difficile* R20291 or the *C. difficile* R20291 Δ*tcdR* mutant ($n = 14$ per strain). Weight and clinical signs of distress were monitored daily. Fecal pellets were collected at 1, 2, and 4 days post-challenge and total *C. difficile* CFUs were enumerated on TCCFA agar; samples were then heat-treated to kill vegetative cells for spore CFU enumeration. Necropsy was performed

2 and 4 days post-challenge, and cecal tissue was harvested for 16S rRNA sequencing ($n = 5$ per group on day 4 post-challenge); cecal tissue RNA was isolated from mice ($n = 3$ per group) on day 2 for gene expression analysis via NanoString.

**Histopathological examination of the mouse cecum and colon.** At the time of necropsy, tissue from the cecum and colon were prepared for histology by placing the intact tissue into histology cassettes and stored in 10% buffered formalin for 48 h at room temperature, then transferred to 70% ethyl alcohol for long term storage. Tissues were processed, paraffin embedded, sectioned at 4 µm thickness, and hematoxylin and eosin stained for histopathological examination (University of North Carolina Animal Histopathology & Lab Medicine core). Histological specimens were randomized and scored in a blinded manner by a board-certified veterinary pathologist (SM). Edema, inflammation (cellular infiltration), and epithelial damage for the cecum and colon were each scored 0–4 based on a previously published numerical scoring scheme[18]. Edema scores were as follows: 0, no edema; (1) mild edema with minimal (2×) multifocal submucosal expansion or a single focus of moderate (2–3×) submucosal expansion; (2) moderate edema with moderate (2–3×) multifocal submucosal expansion; (3) severe edema with severe (3×) multifocal submucosal expansion; (4) same as score 3 with diffuse submucosal expansion. Cellular infiltration scores were as follows: 0, no inflammation; (1) minimal multifocal neutrophilic inflammation of scattered cells that do not form clusters; (2) moderate multifocal neutrophilic inflammation (greater submucosal involvement); (3) severe multifocal to coalescing neutrophilic inflammation (greater submucosal ± mural involvement); (4) same as score 3 with abscesses or extensive mural involvement. Epithelial damage was scored as follows: 0, no epithelial changes; (1) minimal multifocal superficial epithelial damage (vacuolation, apoptotic figures, villus tip attenuation/necrosis); (2) moderate multifocal superficial epithelial damage (vacuolation, apoptotic figures, villus tip attenuation/ necrosis); (3) severe multifocal epithelial damage (same as above) $+/-$ pseudomembrane (intraluminal neutrophils, sloughed epithelium in a fibrinous matrix); (4) same as score 3 with significant pseudomembrane or epithelial ulceration (focal complete loss of epithelium). Photomicrographs were captured on an Olympus BX43 light microscope with a DP27 camera using the cellSens Dimension software.

**RNA extraction from cecal tissue and cecal content.** RNA was extracted from cecal tissue using the PureLink RNA Mini kit (Thermo Fisher, 12183025) following the manufacturer's protocol. The RNA was treated with Turbo DNase (Thermo Fisher, AM2239); the protocol was modified by increasing the amount of enzyme to 5 µl per sample. After 30 min of incubation in a water bath at 37 °C, 2 µl of Turbo DNase enzyme was added to each sample for a further 30 min of incubation. The RNA was then column purified according to the manufacturer's instructions (Zymo, R1019). PCR with primers specific to intron 1 and exon 3 from the mouse β-actin gene were used to screen samples for genomic DNA after DNase treatment.

For extraction of RNA from cecal content for bacterial RNAseq, the content was thawed on ice, then added to 10 mL of TRIzol Reagent (Thermo Fisher, 15596018) in a 15 mL conical. The content was dispersed by vortexing for 10 s, and then was given 20 min on the benchtop to settle. The TRIzol-cecal content mix was then transferred in 1.2 mL aliquots to eight 1.7 mL centrifuge tubes. Three hundred and fifty microliters of chloroform were added, and the tubes were vigorously inverted for 15 s each, after which they were incubated at room temperature for 20 min. The samples were centrifuged at 14,000 rpm at 4 °C for 20 min. The aqueous phase (~650 µl) was then added to 650 µl of isopropanol that had been supplemented with 5 µg/mL glycogen. Samples were vortexed and incubated on ice for 20 min, and then were centrifuged at 4 °C for 30 min. Pellets were washed three times with 70% ethanol, and then dissolved in sterile deionized water. The samples were then treated with Turbo DNase, with the same augmentation of the protocol that was done for the cecal tissue RNA. For reasons that are unclear, the in vivo samples required multiple rounds of Turbo DNase treatment to remove contaminating genomic DNA, resulting in degradation of the RNA in some samples. After column purification, PCR was performed with primers specific to tcdA and rpoC to confirm the removal of genomic DNA.

**RNA extraction from C. difficile cultures in vitro.** Three independent colonies of 630Δerm and tcdR::ermB each were inoculated into 5 ml of TY broth and grown overnight. These were subcultured 1:500 in 5 ml fresh TY and allowed to grow for 18 h at 37 °C in the anaerobic chamber. The cultures were centrifuged and supernatants were decanted. Pellets were dissolved in 1 ml TRIzol Reagent for 20 min on the bench top, after which time 200 µl of chloroform was added and the cultures were vigorously inverted for 15 s and incubated on the benchtop for a further 20 min. The RNA was precipitated as described above, with ~500 µl of the aqueous phase added to 500 µl of isopropanol with glycogen supplementation. The Turbo DNase treatment as described above was performed once, and RNA was confirmed to be free of genomic DNA with the aforementioned primer sets.

**RNA sequencing and transcriptome analysis.** Sequencing of RNA derived from cecal content and in vitro cultures was performed at the Roy J. Carver Biotechnology Center at the University of Illinois at Urbana-Champaign. Ribosomal RNA was removed from the samples using the RiboZero Epidemiology Kit

(Illumina). RNAseq libraries were prepped with the TruSeq Stranded mRNA Sample Prep Kit (Illumina), though poly-A enrichment was omitted. Library quantification was done via qPCR, and the samples were sequenced on one lane for 151 cycles from each end of the fragments on a NovaSeq 6000 using a NovaSeq S4 reagent kit. The FASTQ files were generated and demultiplexed using the bcl2fastq v2.20 Conversion Software (Illumina). Raw paired Illumina reads were imported into Geneious 10.2.6, where adapters and low-quality reads were removed using BBDuk with a Kmer length of 27, a minimum base quality score of 30, and a minimum average read the quality of 30[20]. Reads less than 30 bases in length (and their paired read) were also discarded. The filtered reads were mapped to the C. difficile 630Δerm genome (NCBI accession no. NC_009089.1) using BBMap with a Kmer length of 10 and no other changes to the default settings. Visual inspection of the data indicated that the majority of reads from three wild type and three tcdR samples, each from day 4, mapped to ribosomal RNA genes. These samples were excluded from the analysis, but are included in the SRA submission. The average number of reads that mapped from wild type ($n = 5$) and the tcdR mutant ($n = 6$) from day 2 were 7,395,921 and 22,768,296, respectively; average reads mapped from day 4 for wild type ($n = 3$) and the tcdR mutant ($n = 3$) were 12,959,255 and 4,319,864, respectively. Differential expression analysis was performed using DESeq2 with no changes to the default settings, and genes were considered differentially expression if they had $\pm 1$ log2 fold change and an adjusted $p$-value of $<0.05$[21]. Gene set enrichment analysis of differentially expressed genes was performed using the GSEA-Pro v3 program (http://gseapro.molgenrug.nl) with user-defined cutoff values of $-1$ and 1. Bar plots of enriched Gene Ontology (GO) terms and log2 fold change values of individual transcripts were generated in GraphPad Prism 8.

**Quantitative reverse transcription PCR.** RNA from cecal tissue, cecal content, and in vitro bacterial cultures was used as template in reverse transcription reactions using the Murine Moloney Virus Reverse Transcriptase (NEB M0253S) following the manufacturer's protocol. Genes that were identified as significant and with relatively large differences in expression in the RNAseq and NanoString data sets were chosen for qRT-PCR validation. The resulting cDNA was diluted 1:5 in deionized water and used as template for quantitative PCR with the SsoAdvanced Universal SYBR Green Supermix (Bio-Rad). Quantification of each gene assayed (Table 1 for primers) was performed via standard curve and copy number was determined by comparison to the housekeeping genes tbp (TATA Binding Protein) for host genes and rpoC (RNA polymerase subunit beta) for C. difficile genes.

**NanoString analysis.** RNA from cecal tissue was submitted to the Lineberger Comprehensive Cancer Center Pre-Clinical Genomic Pathology Core at the University of North Carolina at Chapel Hill for quantification of transcripts via NanoString technology[22]. The RNA was hybridized to probes on the Mouse nCounter Immunology Panel, plus custom probes targeting mouse Mmps and Timps. Raw data were imported into the nSolver Advanced Analysis software for data normalization and differential expression analysis. One mouse (challenged with wild type, day 4 post-challenge) was excluded after principal component analysis (PCA) and hierarchical clustering of the data identified it as an outlier with respect to all other samples. The data were normalized and differential expression analysis was performed within the nSolver Advanced Analysis software. Correction for multiple comparisons was performed using the method of Benjamini-Hochberg. Heatmaps of the data were constructed in R using the "pheatmaps" package (https://cran.r-project.org/web/packages/pheatmap/index.html) and volcano plots were constructed in R with the 'EnhancedVolcano' package[23]. Gene set enrichment analysis was performed using the WebGestalt server (http://www.webgestalt.org) with the following changes to the default parameters: the minimum number of genes required for a pathway was lowered to 5 and the False Discovery Rate was adjusted to 0.1. Enriched pathways were visualized in Prism 8. Cecal tissue RNA from mice infected with R20291 and the ΔtcdR mutant was also used in a separate run with the NanoString Mouse nCounter Inflammation panel customized to include code sets for the Mmps and Timps. The data was analyzed identically in the nSolver Advanced program as described above.

**Confocal microscopy.** IMR90 human fibroblasts were cultured in Eagle's Minimum Essential Medium (EMEM) (ATCC, USA) supplemented with 10% fetal bovine serum at 37 °C with 5% $CO_2$. Cells were seeded on glass coverslips in 24-well plates for 3 days, followed by incubation with 0.5 pM of TcdA and TcdB. After 12 h or 15 h, cells were fixed in PBS with 4% paraformaldehyde for 20 min at room temperature and blocked in 10% normal goat serum (Sigma). Collagen was detected using a mix of antibodies against collagen types I, III, and V (Santa Cruz) in a 1:1:1 ratio, and Alexa Fluor 568-conjugated goat anti-mouse secondary antibody (ThermoFisher). Glass coverslips were mounted using VECTASHIELD mounting media with DAPI (Vector Laboratories). Confocal imaging was performed on Zeiss LSM 880 confocal microscope using a 20× or 40× Plan-Apochromat objective lens (numerical aperture of 1.4) and operated with ZEN software (Carl Zeiss, Inc). For image quantification, maximum fluorescent intensity was quantified from five fields of view per coverslip at 20× magnification using ImageJ software.

**C. difficile growth in defined minimal media supplemented with heat-degraded collagen**. Heat-degraded collagen was generated by heating type I collagen (Advanced BioMatrix) at 100 °C for 4 h. Degraded collagen was then concentrated and the pH was adjusted to 7.0. C. difficile was grown in a well-established defined minimal media (CDMM), from which proline was omitted and 0.5 mg of heat-degraded collagen was substituted. The media was passed into the anaerobic chamber and allowed to reduce for 24 h before inoculation. At the same time, individual colonies of C. difficile were inoculated into TY media for overnight growth, after which they were subcultured 1:100 into fresh TY. After 4 h of growth, the cultures were centrifuged in a microcentrifuge in the anaerobic chamber and washed three times in 1 ml sterile PBS, then inoculated 1:500 into the CDMM. Immediately after inoculation, the cultures were serially diluted and plated onto BHI agar for enumeration of C. difficile at 0 h and 24 h time points. Growth was calculated from four independent experiments.

**C. difficile and Bacteroides growth in defined minimal media supplemented with Pro-Gly or Gly-Pro dipeptides, and other collagen degradation substrates**. To assess the ability of C. difficile to acquire the essential amino acid proline from dipeptides, we utilized a well-established defined minimal media (CDMM), from which proline was omitted and Proline-Glycine or Glycine-Proline dipeptides were substituted. The media was passed into the anaerobic chamber and allowed to reduce for 24 h before inoculation. At the same time, individual colonies of C. difficile were inoculated into TY media for overnight growth, after which they were subcultured 1:100 into fresh TY for 4 h. After the 4 h growth, the cultures were centrifuged in a microcentrifuge in the anaerobic chamber and washed three times in 1 ml sterile PBS, then inoculated 1:500 into the CDMM. Immediately after inoculation, the cultures were serially diluted and plated onto BHI agar for enumeration of C. difficile at the 0 h timepoint. Twenty-four hours later, the cultures were serially diluted and plated for CFU enumeration again. Growth was calculated from three independent experiments.

Bacteroides thetaiotaomicron VPI-5482 and Bacteroides fragilis Bf NCTC 9343 were cultured anaerobically at 37 °C from glycerol stocks into tryptone-yeast extract-glucose (TYG) medium and grown overnight[24]. Cultures were back diluted to an OD 600 nm of ~0.1 the next day into minimal media (MM) containing 0.25% glucose, proline, or hydroxyproline[25]. CFUs were enumerated on BHI-blood agar plates at 0 h and after 16 h of growth. Fold change in growth was calculated from 3 independent experiments.

**16S rRNA Illumina sequencing and microbiome analysis**. DNA was isolated from cecal snips at the University of Michigan Microbial Systems Molecular Biology Laboratory. The Mag Attract Power Microbiome kit (Mo Bio Laboratories, Inc.) was used to isolate DNA from cecal snips. Dual-indexing sequencing approach was used to amplify the V4 region of the 16 S rRNA gene. Each PCR mixture contained 2 µl of 10X Accuprime PCR buffer II (Life Technologies, CA, USA), 0.15 µl of Accu-prime high-fidelity polymerase (Life Technologies, CA, USA), 5 µl of a 4.0 µM primer set, 1 µl DNA, and 11.85 µl sterile nuclease free water. The template DNA concentration was 1 to 10 ng/µl for a high bacterial DNA/host DNA ratio. The PCR conditions were as follows: 2 min at 95 °C, followed by 30 cycles of 95 °C for 20 s, 55 °C for 15 s, and 72 °C for 5 min, followed by 72 °C for 10 min. Libraries were normalized using a Life Technologies SequalPrep normalization plate kit as per the manufacturer's instructions for sequential elution. The concentration of the pooled samples was determined using the Kapa Biosystems library quantification kit for Illumina platforms (Kapa Biosystems, MA, USA). Agilent Bioanalyzer high-sensitivity DNA analysis kit (Agilent CA, USA) was used to determine the sizes of the amplicons in the library. The final library consisted of equal molar amounts from each of the plates, normalized to the pooled plate at the lowest concentration. Sequencing was done on the Illumina MiSeq platform, using a MiSeq reagent kit V2 (Ilumina, CA, USA) with 500 cycles according to the manufacturer's instructions, with modifications[26]. Sequencing libraries were prepared according to Illumina's protocol for preparing libraries for sequencing on the MiSeq (Ilumina, CA, USA) for 2 or 4 nM libraries. PhiX and genomes were added in 16S amplicon sequencing to add diversity. Sequencing reagents were prepared according to the Schloss SOP (https://www.mothur.org/wiki/MiSeq_SOP#Getting_started), and custom read 1, read 2 and index primers were added to the reagent cartridge. FASTQ files were generated for paired end reads.

Raw reads were processed in QIIME2, with DADA2 used for de-noising and generating amplicon sequence variants (ASVs)[27,28]. Taxonomic assignment of the ASVs was done using the Silva reference database (silva-132-99-nb-classifier.qza)[29]. The code used to process the reads can be found in Supplementary Data 7 for 630Δerm and Supplementary Data 8 for R20291. Percent relative abundances of Family level ASVs were calculated for each sample in Excel, averaged across treatment groups, and visualized in GraphPad Prism 8.

PCA was performed in the R statistical programming environment (https://www.r-project.org). ASVs from the V4 region that differ by just one base pair may come from the different 16S copies in the same genome[30]. We therefore aggregated ASVs into 99%-identity OTUs using complete-linkage clustering on the Levenshtein edit distances between ASV sequences. PCA was then performed on Hellinger-transformed OTU abundances[31] and the first two principal components plotted with sample scaling (scaling 1). The R code can be found in Supplementary Data 9. The essential R packages used were biomformat, Biostrings, phyloseq,

vegan, ggplot2, and data importing and manipulation packages from the tidyverse package collection[32,33].

**Statistical analysis**. With the exception of the RNAseq analysis, all statistical tests were performed in GraphPad Prism 8. Kruskal–Wallis One-Way ANOVA with Dunn's correction for multiple comparisons was used to test for significance when comparing C. difficile CFUs, mouse weights, cecal toxin activity, and C. difficile collagen growth studies. Histopathology summary scores were tested for significance using a Geissner-Greenhouse corrected ordinary Two-Way ANOVA with Tukey's multiple comparisons test. A mixed effects model with Tukey's multiple comparison's test was used to test for significance on qRT-PCR data. A Mann–Whitney test was used to compare the fluorescence quantification of IMR90 cells treated with vehicle or toxins. A One-way ANOVA with Tukey's multiple comparisons test was used to compare the growth studies with Bacteroides. DESeq2 identified statistically significant differentially expressed genes in the RNAseq study. Differential expression analysis for the NanoString host expression data was performed in the nSolver Advanced software with a Benjamini-Hochberg adjusted p-value. GSEA-Pro v3 uses a hypergeometric testing method to identify overrepresented classes of genes. A p-value of <0.05 was considered statistically significant, with *p < 0.05, **p < 0.01, ***p < 0.001, ****p < 0.0001.

**Reporting summary**. Further information on research design is available in the Nature Research Reporting Summary linked to this article.

## Data availability

Raw sequences have been deposited in the Sequence Read Archive (SRA) with SRA study number SRP252577 and BioProject ID PRJNA612095. Source data are provided within each Supplementary Data file. Other data and biological materials are available from the corresponding author upon reasonable requests.

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

## Acknowledgements

The authors thank Dr. Rita Tamayo for the isogenic *tcdR::ermB* mutant. The Microscopy Services Laboratory, Department of Pathology and Laboratory Medicine, is supported in part by P30 CA016086 Cancer Center Core Support Grant to the UNC Lineberger Comprehensive Cancer Center. J.R.F. and M.H.F. are supported by the University of North Carolina Center for Gastrointestinal Biology and Disease T32DK07737 post-doctoral fellowship and a North Carolina State University College of Veterinary Medicine intramural award. C.M.T. and C.M.P. are funded by the National Institute of General Medical Sciences of the National Institutes of Health under award number R35GM119438.

## Author contributions

J.R.F., C.M.P., R.J.P., A.J.R., and M.H.F. performed the experiments. S.A.M. conducted blinded histology scoring, imaging, and analysis of murine cecal and colonic tissue. J.R.F., C.M.P., M.H.F., and C.M.T. designed the experiments and J.R.F., C.M.P., M.H.F., and M.R.M. analyzed and interpreted the data. J.R.F., C.M.P., M.H.F., M.R.M., and C.M.T. wrote the paper. All authors edited the manuscript.

## Competing interests

C.M.T. consults for Vedanta Biosciences, Inc. and Summit Therapeutics.
