## [Peer Review File · Nature Communications]

REVIEWER COMMENTS

Reviewer #1 (Remarks to the Author):

In their study, Fletcher et al. investigate the role of *C. difficile* toxin production within an antibiotics-treated mouse model. Specifically, differences in the infection between wild-type *C. diff* (WT) and a low toxin-producing derivative (tcdR) are profiled by parallel application of pathogen and host transcriptomics. This analysis revealed toxin production to induce a proinflammatory immune response in the cecal and colonic epithelium, with signs of ECM degradation, and in turn, altered bacterial metabolism between *C. diff* WT and tcdR, where the WT likely exploits amino acids derived from host collagen degradation. Noteworthy, 16S profiling reveals *C. diff* toxin production and host inflammation coinciding with a reduced relative abundance of gut commensals of the Bacteroidaceae family, whose member species seem to otherwise compete with *C. diff* for host-derived (hydroxy)proline.

Overall, I consider this a highly relevant, timely, and – to the most parts – technically well conducted study. I have, however, the following comments that I hope will further improve the manuscript.

- Material & methods (line 217): Are the authors sure that they used the “TruSeq Stranded mRNA Sample Prep Kit (Illumina)” for cDNA library generation? This kit uses oligo-dT beads to capture polyA tails and is thus compatible with eukaryotic but not bacterial RNA-seq (eukaryotic mRNAs possess a polyA tail, which is absent from bacterial mRNA). See this link for details: <https://emea.illumina.com/products/by-type/sequencing-kits/library-prep-kits/truseq-stranded-mrna.html?langsel=/de/>
- Results: Hierarchical clustering of differentially expressed host genes. The authors decide to select the top 50 genes that were differentially expressed between WT-infection and that with the tcdR mutant (based on significance). As a result of this analysis, they obtain two clusters: The WT samples cluster together and the tcdR and uninfected samples form another cluster. This separate clustering of WT and tcdR samples is not surprising given the selection of genes used for this analysis. Could the authors repeat the cluster analysis but without pre-selection of genes (rather do it on all detected or regulated genes)? This would provide a global (and unbiased) view at the data and could be included as an additional supplementary figure.
- Results: Statistical significance. I am not an expert here, but the values derived from statistical significance calculations sometimes seem counterintuitive. For instance, the comparisons in Fig. 4c are not statistically significant according to the authors (see lines 508/509), nor the comparisons

MM-Glucose vs MM-Glucose-Pro or MM-Glucose Hyp in Fig. 4c,d (line 556), nor is the comparison of C9 transcript levels between uninfected and WT-infected mice at 2 days in Suppl. Fig. 6B. Judged by mere eye, however, the differences are quite convincing. Would maybe the addition of a few more replicates help reaching statistical significance?

- Results: A central finding of this study is that *Bacteroides* spp. levels are reduced in the microbiota of mice infected with toxin-proficient *C. diff* and that (at least 2 of these) *Bacteroides* species grow better in presence of glucose and (hydroxy)proline than on glucose alone (Fig. 5c, d), leading to the hypothesis that these commensals compete with *C. diff* for host-derived nutrients. To support this hypothesis more directly, the authors could repeat the in vitro assay (Fig. 5c, d), but this time for co-cultures of *Bacteroides* and either *C. diff* WT or the *tcdR* mutant.

- Discussion: Competition for nutrients between gut commensals and enteric pathogens was also discovered in previous studies. The authors may want to give a few examples to put their own findings in context of the existing literature.

Suggestions for text/figure edits:

- Throughout the text and figures: The authors use different terms for the control mice, such as “no *C. diff*”, “cefoperazone”, “uninfected”, which makes reading quite confusing. Could the authors decide on one expression and use it consistently throughout the manuscript and figures, please?

- Line 50/51: “... inflammation can be beneficial for prominent enteric pathogens such as *Salmonella enterica* and *Vibrio cholera*...” The relevant references (albeit given later on in the Discussion section) should be introduced here already.

- Line 250/250 (and 457-459): Could the authors explain the original rationale to include custom probes for *Mmps* and *Timps* in the NanoString analysis? It seems as if they hypothesized beforehand that those genes would be strongly differentially expressed between WT and *tcdR* (which they indeed turned out to be).

- Line 392: “... CD1917, encoding *eutE*...” I believe the former (CD1917) refers to the gene and the latter (*eutE*) to the gene product. *EutE* should thus be capitalized and non-italic.

- Lines 400-402: The authors report the number of differentially expressed genes for the individual comparisons. It might be easier for the reader to share their excitement if the total number of C. diff genes would be mentioned somewhere.

- Lines 403-405: Could the authors please explain based on what criteria they selected the target genes for qPCR validation? Were those randomly selected from the differentially expressed genes in the RNA-seq data? It rather seems as if those genes were selected because they are representatives for the key pathways reported as differentially enriched in the gene set enrichment analysis. I'd suggest the authors change the order of the respective text passages in the main text; i.e. first introduce the pathways analysis and afterwards describe the qPCR validation (and I'd further suggest the same structure when they discuss the host expression data [lines 469-490]).

- Line 469: "(data not shown)" I'd appreciate if the authors would show these data, even if no genes are significantly differentially expressed in those comparisons. The respective volcano plots could be included in Suppl. Fig. S4. This would maybe also provide some insight into the (quite unusual) finding, that in the pathway enrichment analysis for tcdR day 4 vs tcdR day 2, all pathways were upregulated and not a single one downregulated (see Suppl. Fig. S2b).

- Line 477/478: "... was enriched in transcripts with increased abundance in cecal tissue from wild type mice at both days." Enriched compared to what?

- Lines 524, 527, 532: The acronym "ASV" is used already in lines 524 and 527; but the full term (amplicon sequence variants) is mentioned first in line 532.

- Lines 564-566: The CFU data for the clinical isolate do not recapitulate those of the lab strain (Fig. 1b, c and Suppl. Fig. S1b, c vs Fig. 6a, b). A sentence speculating about possible reasons should be added. Also, for consistency reasons, could the authors plot vegetative cells only in Fig. 6a (like they did for the lab strain in Fig. 1b)?

- Line 573: "Given the pleiotropic nature of the tcdR mutation in the R20291 strain..." Was the pleiotropic nature of this mutation shown in the present study (then, I missed it) or in previous work (in that case, a reference to that paper should be included).

- Figures throughout: Please homogenize the font as much as possible. For example, the same font size should be used for axis labels over different panels of the same figure; as of now, font size varies quite substantially within figures (take Fig. 1 as an example).

- Fig. 1e: Could the authors explain why they detect epithelial damage even in the absence of *C. diff* infection (at 4 days)?

- Fig. 2b: Please label the color bar with its unit (i.e. log₂ fold-change) directly in the figure panel (not only in the caption). Also from this heatmap, the statistical significance of differential gene expression cannot be inferred. Could the authors label genes that reached statistical significance, or – in case all did – mention this in the figure caption? Finally, known CodY-regulated genes are colored. It remains, however, unclear where the information about CodY-regulation stems from. The authors may want to add a sentence explaining this in the figure caption.

- Fig. 3c: I believe the data for Mmp3 is derived from qPCR and the rest from Nanostring. If so, could the authors plot the individual data points (rather than the mean or average) for the different qPCR replicates for Mmp3 (as they do for qPCR data throughout the rest of the figures)?

- Fig. 4a: The font of the scale bar label (“10 μm”) should be increased; as of now it is barely readable. Also, it looks like there are DNA speckles in untreated cells, whereas the DNA stain is more homogeneously distributed over the nuclei of toxin treated cells. Can the authors explain this?

- Fig. 6d: Are the plotted values derived from Nanostring or from qPCR measurements? The caption does not mention that. If this is qPCR data, please plot individual data points from the replicates.

- Suppl. Fig. S6b: Copy number determination of C9 transcripts in WT-infected mice at day 4 reveals two quite distinct clusters, separated by more than one order of magnitude. Can the authors explain this finding? Was a similarly heterogeneous C9 expression also seen in the Nanostring data?

- Suppl. Fig. S6f: Please add the unit label (log₂ fold-change) to the color bar.

Reviewer #2 (Remarks to the Author):

“*Clostridioides difficile* exploits toxin-mediated inflammation to alter the host nutritional landscape and exclude competitors from the gut microbiota”, by J. R. Fletcher, et al., aims to demonstrate that

(1) toxin-mediated inflammation caused by *C. difficile* results in a change in the gut environment that (2) benefits *C. difficile* growth and persistence through (3) changes in nutrient availability and (4) changes in the microbiota composition. Overall, the manuscript was well written and demonstrated interesting findings.

Minor points:

- 1) For the third part of the hypothesis, in vivo results should be strengthened. Can mice given the knockout bacteria be supplemented with proline to show more clearly that this result of inflammation is providing a benefit?
- 2) Are Bacteroidaceae unable to come back because *C. difficile* is already there, using a shared nutrient, or is it due to the inflammation caused by *C. difficile*? This could potentially be addressed by co-administering Bacteroidaceae and *C. difficile*.
- 3) Lines 538-540, the possibility of cage effects should be addressed.
- 4) If the bacteria have access to more nutrients why do the overall bacterial numbers differ by no more than ½ a log.
- 5) Figure lettering is capitalized in figures but lowercase in text.
- 6) Lines 49-51 should have citations.
- 7) Lines 400-402 should have a figure/table reference.
- 8) For histology/immunofluorescent samples, arrows pointing to the various aspects (e.g., epithelial damage) would be helpful, as would indicating what the two colors (red vs. blue) represent in Figure 4A).
- 9) The information in line 514 (“compared to their respective 0 hour time point”) should be in the figure caption.
- 10) Line 633-634 is unclear. Which experiments were conducted a year apart, and what is the significance of this?
- 11) Lines 695-698 N values? Are they the same as in part B?
- 12) Line 716: MFI of the whole well? MFI of selected frames? MFI of representative images?
- 13) Figure 2A: why is there both a red and black bar for structural molecular activity?
- 14) The scales on the two images for Figure 4A are slightly different in size.
- 15) Figure 5C and 5D should have log scales, they are looking at bacterial CFU.
- 16) Figure S3 and S6 need a clearer indication of which samples were used. Was it every sample used in the RNAseq experiment or only a subset? Do dots represent individual samples or technical

repeats? Although it is possible to count the dots to get an N number, listing it in the caption would be appreciated.

17) It would be interesting to perform an FMT with infected feces into an uninfected mouse to show increased spread as a result of the increased spores seen in Figure 6 to show the clinical importance of this, since spores are typically for spreading rather than causing damage. This difference is not discussed.

Reviewer #3 (Remarks to the Author):

This is a fascinating exploration of how *C. difficile* toxin-mediated host inflammation results in changes in bacterial and host metabolism. The innovative hypothesis is that *C. difficile* causes inflammation in the host to promote nutrient acquisition. The authors infect mice with isogenic strains of *C. difficile* that differ only in expression of the *C. difficile* toxins. While these isogenic strains differ in gene expression only in toxin expression during *in vitro* cultivation, when placed into a mouse there are a host of bacterial metabolic pathways that are altered. In the colon of infected mice expression of the toxins and subsequent inflammation resulted in induction of catabolic pathways that are potential sources of nutrients, as well as changes in the bacterial microbiome that could remove competing bacteria, again creating a niche for *C. difficile* to flourish.

One limitation of the hypothesis is that in the absence of toxin-mediated inflammation there is no observable change in the amount of *C. difficile* bacterium excreted in stool (either the vegetative or the spore forms of the bacterium). So I am having trouble envisioning how one can discern or have a read out for a favorable host environment in the absence of an inflammation-induced change in bacterial growth in the host.

I wonder if there are interventions that could be used to test their hypothesis that inflammation creates a favorable environment for *C. difficile*. Could the microbiome be reconstituted to a non-inflamed composition? Or could key metabolic pathways in the host or bacterium be knocked out? These are ambitious experiments that likely fall outside the scope of this manuscript, but interesting to speculate on.

The hypothesis that *C. difficile* induces inflammation to provide a nutrient – rich environment is paradigm shifting and will influence research in this field and in host-microbe studies in general.

We thank the reviewers for their insightful comments. We feel we have addressed the reviewers' comments in this point-by-point response and the revised manuscript is better for it. **Please see authors responses to reviewers' comments in bold below. Line numbers in bold are reflected in the revised manuscript.**

Reviewer 1

- 1) Material & methods (line 217): Are the authors sure that they used the "TruSeq Stranded mRNA Sample Prep Kit (Illumina)" for cDNA library generation? This kit uses oligo-dT beads to capture polyA tails and is thus compatible with eukaryotic but not bacterial RNA-seq (eukaryotic mRNAs possess a polyA tail, which is absent from bacterial mRNA).

We thank the reviewer for catching this detail. This was the kit that was used, however the oligo-dT bead step was omitted. We have revised the manuscript to indicate this now on line 221.

- 2) Results: Hierarchical clustering of differentially expressed host genes. The authors decide to select the top 50 genes that were differentially expressed between WT-infection and that with the *tdcR* mutant (based on significance). As a result of this analysis, they obtain two clusters: The WT samples cluster together and the *tcdR* and uninfected samples from another cluster. This separate clustering of WT and *tcdR* samples is not surprising given the selection of genes used for this analysis. Could the authors repeat the cluster analysis but without pre-selection of genes (rather do it on all detected or regulated genes)? This would provide a global (and unbiased) view at the data and could be included as an additional supplementary figure.

We agree that selecting the top 50 will bias the clustering of samples in the heatmap. We have generated two additional heatmaps for reviewers to look at here: 1) one with all transcripts detected, even those that were unchanged between samples (Figure 1R), and 2) one with every transcript that was called as differentially expressed in either wild type vs. *tcdR* mutant comparison (day 2 and day 4) (Figure 2R). We still feel that these figures would not be helpful to add to supplemental figures as this information is already provided in the Supplemental Files. We felt that providing the raw data to researchers was important so they could mine the data themselves. We continue to think this is a major strength of our work.

- 3) Results: Statistical significance. I am not an expert here, but the values derived from statistical significance calculations sometimes seem counterintuitive. For instance, the comparisons in Fig. 4c are not statistically significant according to the authors (see lines 508/509), nor the comparisons MM-Glucose vs MM-Glucose-Pro or MM-Glucose Hyp in Fig. 4c,d (line 556), nor is the comparison of C9 transcript levels between uninfected and WT-infected mice at 2 days in Suppl. Fig. 6B. Judged by mere eye, however, the differences are quite convincing. Would maybe the addition of a few more replicates help reaching statistical significance?

Thank you for this comment. After going back and looking at all the figures and statistical tests we used, we found that there was a statistical difference in the figures mentioned here. Please see the updated figures and legends in the revised manuscript. The following figures were updated based on the following statistical tests. We also consulted with a statistician to ensure that the tests were appropriate. We have also reflected this change in the results section of the manuscript.

-For Fig. 5c and d we did a One-way ANOVA with Tukey multiple comparison post-test correction. We originally did a Kruskal-Wallis test with Dunn's post-test, and there were limitations with the sample size that make this test unable to detect differences in any magnitude.

-For Fig. 4c we did a more conservative Kruskal-Wallis test with Dunn's multiple comparisons test and found that the pro+ vs. the pro- was statistically different.

-Suppl. Fig. 6b continues to not be significant. Since this is a supplemental figure showing 1 out of 5 genes that we selected to verify our NanoString data we do not feel like adding new replicates to this figure would strengthen this part of our conclusions.

- 4) Results: A central finding of this study is that *Bacteroides* spp. levels are reduced in the microbiota of mice infected with toxin-proficient *C. diff* and that (at least 2 of these) *Bacteroides* species grow better in presence of glucose and (hydroxy)proline than on glucose alone (Fig. 5c, d), leading to the hypothesis that these commensals compete with *C. diff* for host-derived nutrients. To support this hypothesis more

directly, the authors could repeat the *in vitro* assay (Fig. 5c, d), but this time for co-cultures of *Bacteroides* and either *C. diff* WT or the *tcdR* mutant.

We very much agree with the reviewer that the hypothesis should be directly tested in future experiments, and we have spent considerable time trying to formulate an appropriate assay for this manuscript and others. However, there are several major obstacles to performing this proposed experiment that may not be appreciated.

In a rich media that would support the growth of both organisms, we wouldn't be able to test for the competition of proline/hydroxyproline or other nutrients, since both organisms are generalists and can forage for different sources of proline/hydroxyproline-containing molecules and other necessary nutrients. Our lab has found that competition studies with *C. difficile* and individual commensal bacteria in rich media *in vitro* often results in *C. difficile* "winning" the competition (see the recent publication from the Theriot lab, "Strain-Dependent Inhibition of *Clostridioides difficile* by Commensal Clostridia Carrying the Bile Acid-Inducible (*bai*) Operon" by Reed, et al. in *Journal of Bacteriology* for an example). A limitation of these types of competition studies is that they are susceptible to being decided artificially by the organism with the shorter lag and the quicker doubling time *in vitro* rather than truly testing nutrient competition.

We are not aware of a defined minimal medium that will support the growth of both *C. difficile* and either of *B. fragilis* or *B. thetaiotaomicron*, which would be required to address your comment. This is something that we are working on developing currently in the lab. However, even with an appropriate media in hand the results of this assay would be difficult to interpret due to the fact that the *Bacteroides*, which are not auxotrophic for proline/hydroxyproline, may not attempt to acquire these amino acids until late in the *in vitro* competition when *C. difficile* has already sequestered and metabolized them. This idea is supported by other's work that suggests *B. thetaiotaomicron* coordinates the expression of its nutrient acquisition genes to go after preferred nutrients first (PMID 23646867, 26556271), and that it may do so in order to promote microbial co-existence (PMID 29018117).

Additionally, we think that within a microbial community there are multiple individual taxa that contribute a proportion to the emergent property of colonization resistance against *C. difficile* by restricting nutrients in the host environment, which we have found difficult to model *in vitro*.

- 5) Discussion: Competition for nutrients between gut commensals and enteric pathogens was also discovered in previous studies. The authors may want to give a few examples to put their own findings in context of the existing literature.

We thank the reviewer for this suggestion and think that this will strengthen our manuscript. We have added text in the discussion in lines 665-674 to address this, and have included four references highlighting the role of competition for carbohydrates and amino acids in excluding or reducing burdens of other enteric pathogens.

Additional text reads: *In support of their role as potential competitors, a recent report found that a five member cocktail of mucin saccharide metabolizing bacteria that included Akkermansia and Bacteroides was able to reduce the C. difficile burden in a murine model of experimental CDI69. While we focused on proline and hydroxyproline in this study, our RNA seq data suggests that a number of other amino acids and carbohydrate may be more abundant in the C. difficile toxin-inflamed gut. Recent studies have highlighted the role of members of the microbiota in providing colonization resistance to enteric pathogens via competition for both carbohydrate and amino acids70–72. Taken together, our data suggest that C. difficile may pre-emptively compete by remodeling the gut environment to supply nutrients and prevent the return of competitors.*

- 6) Throughout the text and figures: The authors use different terms for the control mice, such as "no *C. diff*", "cefoperazone", "uninfected", which makes reading quite confusing. Could the authors decide on one expression and use it consistently throughout the manuscript and figures, please?

We thank the reviewer for pointing this out and we agree that the use of multiple expressions is confusing. We have decided to use "uninfected or no *C. diff*" mice as we think it is the clearest

and most important descriptor for this group, and have made the appropriate changes throughout the manuscript and figures. We define this group of mice in Fig. 1a and then it is now consistent throughout the manuscript.

- 7) Line 50/51: "... inflammation can be beneficial for prominent enteric pathogens such as Salmonella enterica and Vibrio cholera..." The relevant references (albeit given later on in the Discussion section) should be introduced here already.

We thank the reviewer for catching this. The appropriate references are now in place.

- 8) Line 250/250 (and 457-459): Could the authors explain the original rationale to include custom probes for *Mmps* and *Timps* in the NanoString analysis? It seems as if they hypothesized beforehand that those genes would be strongly differentially expressed between WT and *tcdR* (which they indeed turned out to be).

The reviewer is correct that we hypothesized beforehand that toxin activity would affect expression of multiple *Mmps*. We based our hypothesis off of a publication examining disruption of the actin cytoskeleton and production of activated MMP2 in bovine smooth muscle cells (Koike, T., et al. *Biochem Biophys Res Commun.* 2000). We have added a sentence referencing this in lines 636-639 to address this.

Additional text: *Purified TcdB stimulated MMP2 activity in bovine smooth muscle cells in vitro, suggesting that increased expression and activity of MMPs was be a general consequence of disruption of the actin cytoskeleton by C. difficile toxin activity*⁵⁸.

We performed a small *in vitro* pilot experiment with a murine immortalized small intestinal epithelial cell line and found increased expression of *Mmps* after exposure to either TcdA, TcdB, wild type R20291 overnight supernatants, but not $\Delta tcdR$ supernatants. We had already planned on defining the host transcriptional response to the wild type and mutant, so adding the custom probe set targeting the *Mmps* and *Timps* was an easy supplement.

- 9) Line 392: "... CD1917, encoding eutE..." I believe the former (CD1917) refers to the gene and the latter (eutE) to the gene product. EutE should thus be capitalized and non-italic.

We have capitalized EutE and removed the italics.

- 10) Lines 400-402: The authors report the number of differentially expressed genes for the individual comparisons. It might be easier for the reader to share their excitement if the total number of *C. diff* genes would be mentioned somewhere.

We have added the following, "...of the 3,548 protein coding genes in the *C. difficile* 630 Δ *erm* genome,..." to the sentence in lines 401 to address this.

- 11) Lines 403-405: Could the authors please explain based on what criteria they selected the target genes for qPCR validation? Were those randomly selected from the differentially expressed genes in the RNA-seq data? It rather seems as if those genes were selected because they are representatives for the key pathways reported as differentially enriched in the gene set enrichment analysis. I'd suggest the authors change the order of the respective text passages in the main text; i.e. first introduce the pathways analysis and afterwards describe the qPCR validation (and I'd further suggest the same structure when they discuss the host expression data [lines 469-490]).

The genes selected for qPCR validation were chosen based on patterns of expression in the RNA seq/NanoString data and the pathways analyses. We have moved the qPCR validation sentences to lines 452-454 for the RNA seq, and lines 498-500. We also added our rationale for choosing target genes for validation in the Methods section in lines 245-248 to make this clearer.

Additional text reads: *A subset of genes identified as differentially expressed between wild type and the *tcdR* mutant were selected for quantitative real-time PCR (qRT-PCR) validation, which confirmed trends in expression from the RNAseq (Fig. S3a-d).*

Additional text reads: Genes that were identified as significant and with relatively large differences in expression in the RNAseq and NanoString data sets were chosen for qRT-PCR validation

12) Line 469: “(data not shown)” I’d appreciate if the authors would show these data, even if no genes are significantly differentially expressed in those comparisons. The respective volcano plots could be included in Suppl. Fig. S4. This would maybe also provide some insight into the (quite unusual) finding, that in the pathway enrichment analysis for *tcdR* day 4 vs *tcdR* day 2, all pathways were upregulated and not a single one downregulated (see Suppl. Fig. S2b).

We have already included the volcano plots for the day 4 vs. day 2 comparisons from the NanoString host transcriptome analysis in Suppl. Fig. S4e/f, however, we note that Suppl. Fig. S2b is from the *C. difficile* RNA seq, not NanoString. I think we need more clarity from the reviewer before we can adequately address this comment.

13) Line 477/478: “... was enriched in transcripts with increased abundance in cecal tissue from wild type mice at both days.” Enriched compared to what?

We have added, “...relative to *tcdR* mice.” to the end of the sentence in line 488.

14) Lines 524, 527, 532: The acronym “ASV” is used already in lines 524 and 527; but the full term (amplicon sequence variants) is mentioned first in line 532.

We have moved the full term to line 535 to correct this.

15) Lines 564-566: The CFU data for the clinical isolate do not recapitulate those of the lab strain (Fig. 1b, c and Suppl. Fig. S1b, c vs Fig. 6a, b). A sentence speculating about possible reasons should be added. Also, for consistency reasons, could the authors plot vegetative cells only in Fig. 6a (like they did for the lab strain in Fig. 1b)?

We have added the following additional text, “...While the host and microbiota responses are largely consistent between the strains, we observed that R20291 $\Delta tcdR$ did not have a CFU defect *in vivo*. The R20291 genome encodes the *cdt* binary toxin locus, which is present and presumably functional in both the wild type and the $\Delta tcdR$ mutant, and inflammation induced by the binary toxins may alter the nutrient pool to support increased growth of $\Delta tcdR^{1,2}$.” to lines 605-610 and included relevant references.

Unfortunately, we only plated for total CFUs in this experiment so we do not have vegetative cell CFU data for this.

16) Line 573: “Given the pleiotropic nature of the *tcdR* mutation in the R20291 strain...” Was the pleiotropic nature of this mutation shown in the present study (then, I missed it) or in previous work (in that case, a reference to that paper should be included).

The appropriate reference (Girinathan et al. mSphere 2017) was included in the sentence in line 580, however, we did not explicitly state that the R20291 *tcdR* mutant was pleiotropic. We have corrected that by including the word when describing the mutant’s sporulation defect in line 579.

17) Figures throughout: Please homogenize the font as much as possible. For example, the same font size should be used for axis labels over different panels of the same figure; as of now, font size varies quite substantially within figures (take Fig. 1 as an example).

We thank the reviewer for this comment and have substantially edited the figures to make the fonts more homogeneous throughout.

18) Fig. 1e: Could the authors explain why they detect epithelial damage even in the absence of *C. diff* infection (at 4 days)?

It has been previously reported that very low levels of histopathological changes occur in the cefoperazone model of CDI (Theriot, C.M., et al. Gut Microbes 2011), and antibiotic treatment can have effects on the host transcriptome independent of the presence of microbes (Morgun, A., et al. Gut 2016). It is also known that antibiotic treatment often leads to an increase in inflammatory Proteobacteria in the gut (McDonald, L.C. Journal of Travel Medicine 2017).

19) Fig. 2b: Please label the color bar with its unit (i.e. log₂ fold-change) directly in the figure panel (not only in the caption). Also from this heatmap, the statistical significance of differential gene expression cannot be inferred. Could the authors label genes that reached statistical significance, or – in case all did – mention this in the figure caption? Finally, known CodY-regulated genes are colored. It remains, however, unclear where the information about CodY-regulation stems from. The authors may want to add a sentence explaining this in the figure caption.

We have added log₂ fold change to the figures as the reviewer suggested. Most genes in the heatmap were significant on at least one day, but not necessarily on both days. Some non-significant genes were included given their presence in an operon that had significant genes. We have provided all of the raw data in the Supplemental File 3 for the reader.

The sentence from lines 730-732 has been amended to clarify the CodY regulation thusly, “The labels of known CodY regulated transcripts (according to Dineen, et al. J Bacteriology 2010) are color-coded in red if they increased in expression in a *codY* mutant *in vitro* and green if they decreased³⁹.”

20) Fig. 3c: I believe the data for Mmp3 is derived from qPCR and the rest from Nanostring. If so, could the authors plot the individual data points (rather than the mean or average) for the different qPCR replicates for Mmp3 (as they do for qPCR data throughout the rest of the figures)?

The reviewer is correct that the Mmp3 data originates from qPCR rather than NanoString. The individual data points from the qPCR are already shown in Fig. S6c.

We have added, “..., and individual data points can be seen in supplemental Figure S6c.” to the end of the sentence in the Fig. 3 legend in lines 742-743 to make this clearer.

21) Fig. 4a: The font of the scale bar label (“10 μm”) should be increased; as of now it is barely readable. Also, it looks like there are DNA speckles in untreated cells, whereas the DNA stain is more homogeneously distributed over the nuclei of toxin treated cells. Can the authors explain this?

Thank you for the suggestion. The font of the scale bar label has been increased. We unintentionally included an image that showed nuclear speckles in the untreated cells, however, this image is not an accurate representation of all the cells. To clarify this, we have changed our image in Fig 4a to one that is more representative of the nuclear morphology.

22) Fig. 6d: Are the plotted values derived from Nanostring or from qPCR measurements? The caption does not mention that. If this is qPCR data, please plot individual data points from the replicates.

We thank the reviewer for catching this. It is NanoString data, and we have added the following, “...derived from NanoString transcriptome analysis.” to the end of the sentence on line 778 in the Fig. 6 legend.

23) Suppl. Fig. S6b: Copy number determination of C9 transcripts in WT-infected mice at day 4 reveals two quite distinct clusters, separated by more than one order of magnitude. Can the authors explain this finding? Was a similarly heterogeneous C9 expression also seen in the Nanostring data?

Exploratory NanoString data showed that expression from one wild type sample from day 4 is an order of magnitude higher; this sample was excluded as an outlier in the analyses presented in the paper, as many genes were relatively highly expressed in this sample compared to all other samples. The outlier had the effect of making our differential expression analysis look considerably more dramatic/significant than it actually was. We have included a heatmap (Figure 3R) and a principal coordinates analysis plot (Figure 4R) including this sample to show the reviewer how different this sample was from all others. It’s exclusion was overlooked when performing the qPCR validation experiments. It is unclear why a second sample exhibits such a difference in expression from the group mean.

24) Suppl. Fig. S6f: Please add the unit label (log₂ fold-change) to the color bar.

We have added log₂ fold change to the figures as the reviewer suggested.

- 1) For the third part of the hypothesis, *in vivo* results should be strengthened. Can mice given the knockout bacteria be supplemented with proline to show more clearly that this result of inflammation is providing a benefit?
We thank the reviewer for their comments and appreciate their suggestions; we have had past discussions about supplementing mice with excess proline to examine how this affects *C. difficile* colonization and disease. We decided against these studies, as we think that digested mouse chow in the mouse gut likely provides sufficient proline and other nutrients to support *C. difficile* growth, especially in mice challenged with the *tcdR* mutant, where inflammation is minimal. As such, we feel that supplementing with proline will not have a major effect on growth of the mutant *in vivo*.
- 2) Are Bacteroidaceae unable to come back because *C. difficile* is already there, using a shared nutrient, or is it due to the inflammation caused by *C. difficile*? This could potentially be addressed by co-administering Bacteroidaceae and *C. difficile*.
This is an excellent question and we don't definitively know the answer yet. That some of the mice challenged with either *tcdR* mutant had both *C. difficile* and Bacteroidaceae in their cecal content suggests that, at least in the absence of inflammation, the nutritional environment is such that *C. difficile* isn't using any nutrient(s) that are essential to the Bacteroidaceae. This suggests that inflammation may be necessary to the prevention/reduction of the return of the Bacteroidaceae in the ceca of wild type mice. We agree that further *in vivo* dissection of this relationship is worth pursuing, but since our aim was to broadly examine the effect of toxin-induced inflammation on multiple facets of the host-pathogen-microbiome axis, we feel that this is outside of the scope of this work. Additionally, co-administration has been performed in reference 67 and the outcome was consistent with our hypothesis.
- 3) Lines 538-540, the possibility of cage effects should be addressed.
We thank the reviewer for pointing this out. We have added the following sentence to lines 540-541: "The Bacteroidaceae were detected in samples from the same cages over time, suggesting a potential cage effect."
- 4) If the bacteria have access to more nutrients why do the overall bacterial numbers differ by no more than ½ a log.
We don't believe that the wild type has access to more nutrients, but rather that it has access to different nutrients than the *tcdR* mutant. As mentioned in our first response, we think that *C. difficile* grows well in digested mouse chow, which is abundant in the gut, especially in mice not experiencing much inflammation. Additionally, we think that the benefit of toxin activity re: nutrients is likely seen at the level of persistence and niche maintenance by resupplying Stickland substrates over the course of a chronic infection, rather than colonization ability. While the cefoperazone mouse model of CDI recapitulates many aspects of CDI in humans, it unfortunately does not model the chronic CDI that many humans experience. Perhaps more importantly, mice are coprophagic, so they will re-inoculate themselves multiple times daily as they consume feces containing high levels of *C. difficile* spores.
- 5) Figure lettering is capitalized in figures but lowercase in text.
We thank the reviewer for catching this detail. This has been corrected in the figures.
- 6) Lines 49-51 should have citations.
References have now been added to this.
- 7) Lines 400-402 should have a figure/table reference.
We have added the supplemental file reference here.
- 8) For histology/immunofluorescent samples, arrows pointing to the various aspects (e.g., epithelial damage) would be helpful, as would indicating what the two colors (red vs. blue) represent in Figure 4A).
Arrows have been added to indicate epithelial damage in Fig. 1f. The color difference in the immunofluorescence images has been added to Fig. 4a.

9) The information in line 514 (“compared to their respective 0 hour time point”) should be in the figure caption.

We have added the relevant information in the figure legend in line 756.

10) Line 633-634 is unclear. Which experiments were conducted a year apart, and what is the significance of this?

The *in vivo* infection models were performed a year apart. We were trying to convey that the negative association between toxin-induced inflammation and Bacteroidaceae was reproducible in two experiments separated by time, and thus was likely to be a true biological phenomenon. We have removed the sentence to as to avoid causing confusion.

11) Lines 695-698 N values? Are they the same as in part B?

Yes, the n values are the same as in part B.

12) Line 716: MFI of the whole well? MFI of selected frames? MFI of representative images?

We quantified the MFI of fields of view at 20x magnification. We have added this information to the methods section.

13) Figure 2A: why is there both a red and black bar for structural molecular activity?

Some genes belonging to that category were decreased in expression, while some increased. The red and black color denotes which one. This is included in the figure legend.

14) The scales on the two images for Figure 4A are slightly different in size.

The reviewer is correct that the scale bars are different sizes. The scale bars were generated by Zen software and not manually. The images in Fig. 4a were not used for quantification. We focused on highlighting the variances in collagen structure in Fig. 4a. While a 40x objective was used for all images in Fig. 4a, the field of view was slightly altered during imaging. In Fig. 4b, new images were acquired using a 20x objective and all parameters were kept constant during imaging and images did not include DAPI.

15) Figure 5C and 5D should have log scales, they are looking at bacterial CFU.

We originally had this figure in log scale, but the difference between MM-Glucose and MM-Glucose+Pro/Hyp is about 2-fold, which gets obscured. We have supplied log scale graphs for the reviewer to see and we will change it if that is still preferred.

16) Figure S3 and S6 need a clearer indication of which samples were used. Was it every sample used in the RNAseq experiment or only a subset? Do dots represent individual samples or technical repeats? Although it is possible to count the dots to get an N number, listing it in the caption would be appreciated. **For both Figures S3 and S6 every dot represents a biological replicate, i.e. RNA from an individual mouse cecum or cecal content. For the S3 qPCR validation, we included all samples, even those deemed unfit for RNA seq due to RNA degradation. As such, some samples failed to amplify some transcripts, leading to variability in the number of dots. For S6 qPCRs, these match the NanoString Ns. We have added the Ns in the figure legends of both figures.**

17) It would be interesting to perform an FMT with infected feces into an uninfected mouse to show increased spread as a result of the increased spores seen in Figure 6 to show the clinical importance of this, since spores are typically for spreading rather than causing damage. This difference is not discussed.

We agree that the strain-specific sporulation defect of the R20291 $\Delta tcdR$ mutant is quite interesting, however, mice are coprophagic, so it is hard to control for this in the study of spore spreading given that mice are constantly re-inoculating themselves. In previous experiments in our lab we have observed that inoculum dosages of 100-fold less than were used in this manuscript do not result in changes in colonization or disease. We do mention that our *in vivo* results are consistent with *in vitro* data from a previous publication (Girinathan, B.P., et al. mSphere 2017), but did not discuss it further as it was neither a novel phenotype nor was sporulation the focus of the manuscript.

Reviewer 3

1) One limitation of the hypothesis is that in the absence of toxin-mediated inflammation there is no observable change in the amount of *C. difficile* bacterium excreted in stool (either the vegetative or the spore forms of the bacterium). So I am having trouble envisioning how one can discern or have a read out for a favorable host environment in the absence of an inflammation-induced change in bacterial growth in the host.

We did observe a small but significant difference in *tcdR* mutant vegetative cell CFUs in the feces at one day post challenge (Fig. 1b, $p < 0.05$), but we understand and agree with the sentiment of this comment. We don't believe the fitness benefit of inflammation is at the level of colonization, rather, it is likely persistence and niche maintenance by resupplying Stickland substrates and excluding competitors/communities that mediate colonization resistance. Unfortunately, the mouse model of CDI is an acute infection, and does not accurately recapitulate a chronic CDI in humans, so that is a limitation of our study. Additionally, *C. difficile* grows well in digested mouse chow, and mice are coprophagic and are constantly re-inoculating themselves with spores in the feces, which would be detected by our plating. This aspect of mouse behavior is hard to control for in this model.

2) I wonder if there are interventions that could be used to test their hypothesis that inflammation creates a favorable environment for *C. difficile*. Could the microbiome be reconstituted to a non-inflamed composition? Or could key metabolic pathways in the host or bacterium be knocked out? These are ambitious experiments that likely fall outside the scope of this manuscript, but interesting to speculate on.

We thank the reviewer for these comments and questions. There are studies in the literature showing that prior inflammation can alter experimental CDI in mice, and we have included a sentence with the appropriate references in lines 642-644. As for reconstituting the microbiome, yes, this has been done in the form of fecal microbiota transplants, and they are highly effective at treating CDI in mice and in humans (Seekatz, A.M., et al. Infection and Immunity 2015). Regarding knocking out key metabolic pathways in *C. difficile*, work is underway in the Theriot lab to address this, though we agree with the reviewer that it is outside the scope of this manuscript.

Figure 1R

Figure 2R

Figure 3R

Figure 4R

REVIEWERS' COMMENTS

Reviewer #1 (Remarks to the Author):

In the revised version of this manuscript, the authors addressed almost all of my previous comments. Overall, I still think this is a timely, relevant and well-conducted study.

In regard of my previous comment #2: Thank you for compiling these heat maps. I'd still appreciate if they would be added to the manuscript (as Suppl. Figure). For example, from these heat maps it is more obvious that some samples are "outliers" and don't cluster with the remaining replicates for the respective condition; this is, however, less obvious in the heat map in Fig. 3a, where the restriction to the 50 top differentially expressed genes between WT and tcdR infection overwhelms this effect (except for a single WT sample). I believe it would be helpful to readers to appreciate this, even without digging into the raw data provided in the Supplemental Files. Also (with respect to the author's response to my previous comment #23): I assume the outlier that the authors refer to in Figs. R3 + R4 is the same that clusters apart from the remaining WT samples in Fig. R2. If I understand the authors correctly, some of the RNA from the same animal was used for qRT-PCR analysis (e.g. Suppl. Fig. S6). The authors should clearly label the data points in the qRT-PCR plots that were derived from this "outlier" sample.

Regarding my previous comment #12 that was unclear to the authors: I am aware that you did show a volcano plot of the comparison "tcdR day 4 vs. tcdR day 2" in Suppl. Fig. S4f. However, I was asking for volcano plots for the comparisons "tcdR vs. no C. diff, day 2" and "tcdR vs. no C. diff, day 4". These are the comparisons you refer to in lines 493-495.

Reviewer #2 (Remarks to the Author):

The authors have addressed my suggestions regarding the previous version of the manuscript. Overall, this is a very interesting contribution that will generate considerable interest in the field.

Reviewer #3 (Remarks to the Author):

The authors have responded to the concerns of the prior critique comprehensively and the revised manuscript is improved as a result.

We thank the reviewers again for their insightful comments. We feel we have addressed reviewer 1's comments in this point-by-point response and in the revised manuscript. **Please see authors responses to reviewers' comments in bold below. Line numbers in bold are reflected in the revised manuscript.**

Reviewer 1

1) In regard of my previous comment #2: Thank you for compiling these heat maps. I'd still appreciate if they would be added to the manuscript (as Suppl. Figure). For example, from these heat maps it is more obvious that some samples are "outliers" and don't cluster with the remaining replicates for the respective condition; this is, however, less obvious in the heat map in Fig. 3a, where the restriction to the 50 top differentially expressed genes between WT and tcdR infection overwhelms this effect (except for a single WT sample). I believe it would be helpful to readers to appreciate this, even without digging into the raw data provided in the Supplemental Files.

We agree with the reviewer that inclusion of the heatmaps as supplemental figures will give the reader an appreciation for how patterns in overall gene expression affect clustering of the groups. We have included them in the manuscript as Fig. S7 and S8.

2) Also (with respect to the author's response to my previous comment #23): I assume the outlier that the authors refer to in Figs. R3 + R4 is the same that clusters apart from the remaining WT samples in Fig. R2. If I understand the authors correctly, some of the RNA from the same animal was used for qRT-PCR analysis (e.g. Suppl. Fig. S6). The authors should clearly label the data points in the qRT-PCR plots that were derived from this "outlier" sample.

The outlier sample was not included in Fig. R2, so the outlying sample the reviewer is referencing here is a different sample. The reviewer is correct, though, that the outlier seen in R3 and R4 was unfortunately included in the qRT-PCR validation of C9 and Mmp12. We have made the data points in each graph a different symbol (open circle) to reflect this, and amended the text in the Fig. S6 legend.

***Additional text reads:** For S6b and S6d, a sample that we had identified as an outlier was inadvertently included in the qRT-PCR validation of C9 and Mmp12 expression levels. We have indicated the data point in each graph derived from this sample by changing the symbol to an open circle.*

3) Regarding my previous comment #12 that was unclear to the authors: I am aware that you did show a volcano plot of the comparison "tcdR day 4 vs. tcdR day 2" in Suppl. Fig. S4f. However, I was asking for volcano plots for the comparisons "tcdR vs. no C. diff, day 2" and "tcdR vs. no C. diff, day 4". These are the comparisons you refer to in lines 493-495.

We thank the reviewer for clarifying the question. We have added the volcano plots for these two comparisons to Fig. S4g and h.